# UncertainSAM: Fast and Efficient Uncertainty Quantification of the Segment Anything Model

Timo Kaiser [1]   Thomas Norrenbrock [1]   Bodo Rosenhahn [1]

## Abstract

The introduction of the *Segment Anything Model* (SAM) has paved the way for numerous semantic segmentation applications. For several tasks, quantifying the uncertainty of SAM is of particular interest. However, the ambiguous nature of the class-agnostic foundation model SAM challenges current uncertainty quantification (UQ) approaches. This paper presents a theoretically motivated uncertainty quantification model based on a Bayesian entropy formulation jointly respecting aleatoric, epistemic, and the newly introduced task uncertainty. We use this formulation to train USAM, a lightweight post-hoc UQ method. Our model traces the root of uncertainty back to under-parameterised models, insufficient prompts or image ambiguities. Our proposed deterministic USAM demonstrates superior predictive capabilities on the SA-V, MOSE, ADE20k, DAVIS, and COCO datasets, offering a computationally cheap and easy-to-use UQ alternative that can support user-prompting, enhance semi-supervised pipelines, or balance the tradeoff between accuracy and cost efficiency.

## 1. Introduction

In the last years, significant effort and resources have been spent into the development of foundation models. One well-established model is the *Segment Anything Model* (SAM) (Kirillov et al., 2023) that segments arbitrary objects in images based on multi-modal prompts. SAM enables researchers and engineers to rapidly develop new applications (Zhang et al., 2023a) such as interactive segmentation (Wu & Xu, 2024) or satellite imagery (Ren et al., 2024) without the need for expensive data annotation or

---

[1]Institute for Information Processing / L3S - Leibniz University Hannover, Germany. Correspondence to: Timo Kaiser <kaiser@tnt.uni-hannover.de>.

*Proceedings of the 42$^{nd}$ International Conference on Machine Learning*, Vancouver, Canada. PMLR 267, 2025. Copyright 2025 by the author(s).

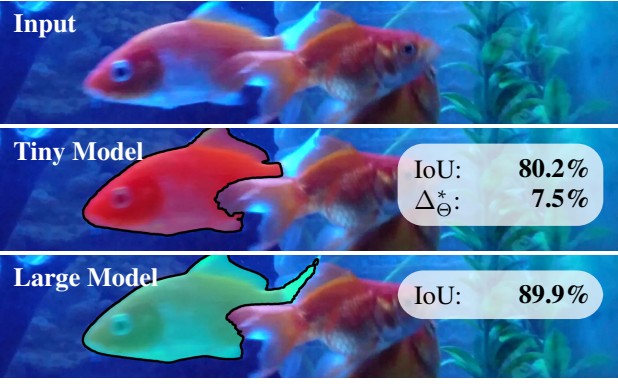

*Figure 1. Large* and *Tiny* SAM with our epistemic UQ on the DAVIS dataset. The *Large* model predicts an accurate mask while the *Tiny* model fails in the area of the tail. Our USAM MLP $\Delta_\theta^*$ estimates a potential epistemic gap of $7.5\%$ intersection over union (IoU) directly from the *Tiny* model, helping the user to balance the trade-off between efficiency and accuracy.

model training. Similarly, the adoption of deep learning in critical domains such as medical diagnosis (Wang et al., 2019) or autonomous driving (Yan et al., 2024a; Gottwald et al., 2024) intensified research in uncertainty quantification (UQ) to make systems more reliable and safe. Moreover, UQ helps to improve model training (Kaiser et al., 2022), supervisability (Vujasinović et al., 2024), and in other applications (Paul et al., 2024; Kaiser et al., 2024; Wehrbein et al., 2025; Kruse & Rosenhahn, 2025). UQ is therefore also moving into focus with regard to SAM (Zhang et al., 2023b; Jiang et al., 2024). Like other segmentation methods, SAM exhibits uncertainty caused by various factors. For example, Figure 1 illustrates the prediction of SAM using tiny and large backbones. While the tiny model is more cost-efficient, it fails to segment the details on the fishtail. Knowing about potential accuracy gaps (*i.e.* uncertainty) based on under-parameterization can help determine if a switch to a larger model is worthwhile. Existing applications rely on SAM's inherent confidence score or apply standard approaches like test-time augmentation (Deng et al., 2023) which are resource-intensive and poorly suited because SAM is subject to a unique kind of uncertainty. In addition to model (epistemic) and data uncertainty (aleatoric), the task is undefined, *i.e.* the object to segment is unknown *a priori*, and the user defines the prior with a prompt. Thus,

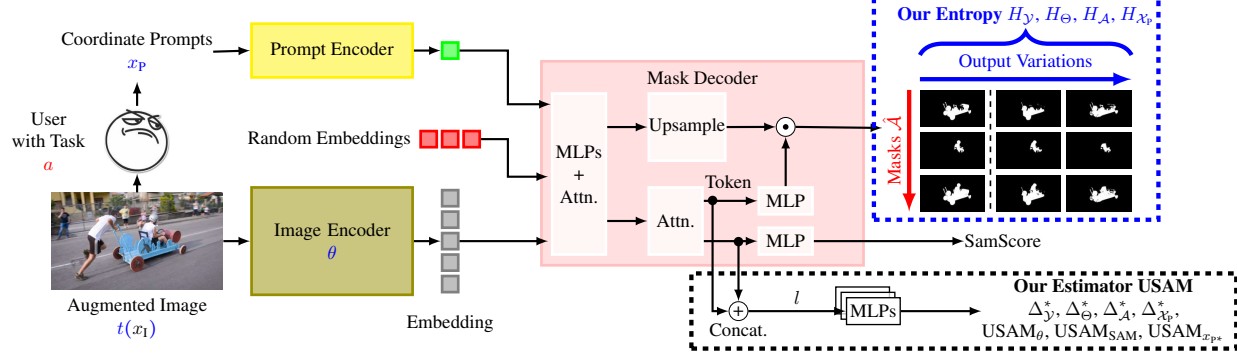

*Figure 2.* The SAM framework with our USAM extension and Bayesian entropy approximation to quantify uncertainty. Starting from the the bottom left, an image $x_I$ is the input. A user defines one or more coordinate prompts $x_P$ to specify his desired segmentation task $a$. The image and prompt are encoded into embeddings, concatenated together with random embeddings, and fed into the mask decoder which applies attention, MLPs and upsampling layers. The SAM framework estimates three potential masks that address different tasks $\hat{A}$. Additionally, SAM estimates the IoU between the ground truth of $a$ and the masks corresponding to $\hat{A}$, denoted as *SamScore*. We extract and concatenate the mask and confidence tokens (Kirillov et al., 2023) to train MLPs that estimate the expected predictive, epistemic, task, and prompt uncertainty. Furthermore, the process to calculate the entropy of uncertainty is visualized in blue. Multiple prompts are chosen by the user, augmentations $t$ are applied to the input, and models of different size $\theta \in \Theta$ are used to apply variational inference.

SAM also underlies task uncertainty because the prompt is potentially describing multiple valid tasks or SAM fails to decode it.

To advance UQ research for SAM and better classify the sources of uncertainty, this paper:

- Introduces a **complete UQ formulation** for SAM,
- Presents its **Bayesian UQ approximation**, and
- Introduces **USAM, an estimator** that outperforms existing methods in estimating sources of uncertainty.

Our **formulation of UQ in SAM** divides the predictive uncertainty into uncertainties stemming from under-parameterized models (epistemic uncertainty), insufficient image prompts (aleatoric prompt uncertainty), or ambiguous images (aleatoric task uncertainty). The **Bayesian approximation** employs Monte Carlo sampling (Mooney, 1997) to quantify those. Since the Bayesian formulation is an important basis for our investigations, but unfortunately computationally expensive, we introduce UncertainSAM (USAM), a simple training strategy to train a lightweight estimator that accounts for the same uncertainties. USAM uses multi-layer perceptrons (MLPs) to directly estimate uncertainties from SAM's pretrained latent representations, enabling practical use without bells and whistles in arbitrary applications. Our experimental results demonstrate that our UQ methods accurately estimate if a prompt should be refined, supervision is required, or larger models improve the prediction. The last application, combined with the efficiency of our MLPs, can reduce the energy requirements of applications. While the computational expensive Bayesian approximation is theoretically grounded, USAM is also relevant for practical usage. Notably, the simple MLPs of USAM perform on par or surpass the Bayesian approximation and existing UQ methods, establishing a new state-of-the-art in UQ for SAM. **Easy-to-use code is available here[1]**.

Section 2 recapitulates SAM, UQ, and UQ in SAM. Then, Section 3 introduces the Bayesian approximation and our novel USAM to estimate uncertainty. Section 4 shows in-depth experimental evaluation concluded in Section 5.

## 2. Segment Anything and Uncertainty

This section offers preliminaries and related work to SAM and UQ. Then, we present a new Bayesian formulation of UQ in SAM that is used later in our method.

### 2.1. Segment Anything Model (SAM)

SAM (Kirillov et al., 2023) is a foundation model for class-agnostic image segmentation. It predicts image masks for arbitrary objects based on an input image $x_I$ and one or more multi-modal input prompts $x_P$. An overview, extended with our contributions, is presented in Figure 2. Formally, SAM estimates the pixel-wise probability of being foreground $y$ for an unknown ground truth $\overline{y}$. The framework comprises three basic blocks: The image and prompt encoders as well as the mask decoder. A Vision Transformer (Dosovitskiy et al., 2021) is used to encode the input image, while multi-modal prompt encoders encode prompts. Prompts can be point or box coordinates encoded as positional encoding (Tancik et al., 2020), text encoded with CLIP (Radford et al., 2021), or dense masks added to the image embedding with convolutions. The resulting embeddings are combined

---

[1]https://github.com/GreenAutoML4FAS/UncertainSAM

with a randomly chosen embedding and fed into the mask decoder which updates and fuses all prompts. The resulting new embedding $l$ is split into a mask and IoU token. The mask token is projected with multi-layer perceptrons (MLP) and multiplied with the upsampled image embedding to create pixel-wise foreground probabilities, *i.e.* the object mask. The IoU token is fed into another MLP that estimates the intersection over union (IoU) (Jaccard, 1901) to the ground truth, denoted as *SamScore* in this paper.

Since the ground truth mask $\overline{y}$ may be unknown even to the user encoding it into the prompt, we omit the notion of ground truth and instead define the user-encoded proposition as the task $a$. Depending on the image $x_I$, multiple prompts $\{x_{P,1}, x_{P,2}, \ldots\}$ can represent the same task $a$ and multiple tasks $\{a_1, a_2, \ldots\}$ can be represented by the same prompt $x_P$, *e.g.* different granularities in hierarchical structures. To tackle this task ambiguity, SAM adds the random embedding to the latent. With multiple random embeddings, multiple masks based on different task assumptions $\hat{a} \in \hat{\mathcal{A}}$ are generated. Selecting the closest to ground truth mask and using the minimum loss (Guzmán-rivera et al., 2012; Charpiat et al., 2008) during training induces variability in the network that only depends on the random embedding. During test time, the mask with highest *SamScore* is kept.

## 2.2. Uncertainty Quantification (UQ)

According to (Gawlikowski et al., 2023), a neural network (NN) parameterized with weights $\theta$ from the function space $\Theta$ maps a set of inputs to a set of targets. The finite training set $\mathcal{D}$ is used to condition $\theta$ during optimization, enabling the NN to predict targets for new samples. UQ aims to quantify the confidence that the prediction is correct.

From a Bayesian perspective (Gal & Ghahramani, 2015), the predictive uncertainty distribution is defined as

$$\underbrace{p(y|x, \mathcal{D})}_{\text{Predictive Uncertainty}} = \int \underbrace{p(y|x, \theta)}_{\text{Aleatoric}} \underbrace{p(\theta|\mathcal{D})}_{\text{Epistemic}} \, d\theta \qquad (1)$$

over all parameter configurations $\theta \in \Theta$ conditioned on the unseen input sample $x$ and the training data $\mathcal{D}$. The formulation allows to decompose uncertainty into more fine-grained aspects, namely the *Aleatoric* uncertainty inherent in $x$ and the *Epistemic* uncertainty that originates from imperfect $\theta$. The posterior distribution $p(\theta|\mathcal{D})$ over all $\theta$ is infeasible, and $p(y|x, \theta)$ is often undefined. Thus, many methods approximate both terms using Monte Carlo simulation (MC) (Mooney, 1997).

The epistemic posterior distribution $p(\theta|\mathcal{D})$ can be approximated with MC samples $\theta \in \Theta$. For example, model ensembles (Lakshminarayanan et al., 2017) reduce Equation (1) to a tractable sum of a few models, while cost intensive Bayesian neural networks (BNN) (Goan & Fookes, 2020) sample from an approximation of $\Theta$. Efficient BNN approaches (Gal & Ghahramani, 2016) assume $\Theta$ to be Bernoulli-distributed modeled with Dropout (Srivastava et al., 2014). The aleatoric uncertainty $p(y|x, \theta)$ is sometimes undefined or biased. Variational inference can approximate it applying test-time augmentations (Wang et al., 2019), where the input measurement $x$ is slightly perturbed (*e.g.* with flips and rotations) with static $\theta$. This approach assumes that target values $\overline{y}$ of ambiguous samples are over-fitted on sample-specific biases during training and only memorized by spurious patterns. Perturbations activate different biases, approximating the distribution of $y$.

For convenience, some methods (Lahlou et al., 2023; Quillent et al., 2024) quantify uncertainty with the variance or entropy of the sampled distribution. This allows deterministic approximations of the epistemic, aleatoric, or predictive variance/entropy. For example, DEUP (Lahlou et al., 2023) trains a NN to estimate the epistemic variance of MC sampling, Compensation learning (Kaiser et al., 2023) predicts pixel-wise aleatoric uncertainty, or Quillent et al. (2024) spatial aleatoric uncertainty. Deterministic UQ methods are lightweight, applicable post-hoc, and efficient.

While the presented methods work well in practice, the theoretical foundations for separating aleatoric and epistemic uncertainty remain debatable and under active discussion. The task-agnostic nature of SAM further amplifies this challenge, as discussed next. More insights into uncertainty can be found in (Gawlikowski et al., 2023; Gruber et al., 2023).

## 2.3. UQ in SAM

SAM inherently faces challenges related to uncertainty, as ambiguities are present in images or prompts. Quantifying the reliability of SAM's prediction is critical, especially in sensitive domains. For instance, in the medical domain, SAM exhibits excessive uncertainty, necessitating Yan et al. (2024b) to retrain SAM or Jiang et al. (2024) to fine-tune the ability to predict different masks. Existing works have introduced or utilized UQ methods in SAM: Deng et al. (2023) or Zhang et al. (2023b) approximate aleatoric uncertainty with the entropy of MC simulations by augmenting prompts, (Vujasinović et al., 2024) calculate the mean pixel-wise entropy between different object masks to detect errors, or (Xu et al., 2024) quantify uncertainty with subjective logic (Jsang, 2018) and use the uncertainty to refine prompts. (Li et al., 2024) adapt UQ in SAM to sample more diverse predictions and (Liu et al., 2024) use entropy to improve finetuning.

Despite these efforts, the implications of SAM's class- and task-agnostic framework on UQ remain underexplored. Unlike other frameworks, the task is undefined *a priori* and SAM is trained to cover a large set of tasks $\mathcal{A}$. During inference, the abstract task $a \in \mathcal{A}$ is defined by the user

and represented by a prompt, *e.g.* a persons that should be segmented is encoded by a single point coordinate. Decomposing SAM's input into the image $x_\text{I}$ and the prompt $x_\text{P}$, and explicitly including the subtask of estimating the task, the Bayesian formulation from Equation (1) extends to

$$
\underbrace{p(y|x_\text{I}, \mathcal{D}, a)}_{\text{Predictive Unc.}} = \int \int \int \underbrace{p(y|x_\text{I}, \hat{a}, \theta)}_{\text{Aleatoric}}
$$
$$
\underbrace{p(\hat{a}|x_\text{I}, x_\text{P}, \theta)}_{\text{Prompt Decoder}} \underbrace{p(x_\text{P}|x_\text{I}, a)}_{\text{Prompt Encoder}} \underbrace{p(\theta|\mathcal{D})}_{\text{Epistemic}} \, \mathrm{d}\theta \mathrm{d}x_\text{P} \mathrm{d}\hat{a} \; . \quad (2)
$$

In this formulation, SAM estimates the task probability $p(\hat{a}|x_\text{I}, x_\text{P}, \theta)$ given the human encoded prompt $x_\text{P}$. This introduces new potential sources of uncertainty: Ambiguity in prompt en- and decoding. The first stems from the user inspecting the image content, the latter from SAM's interpretation of $x_\text{I}$ and $x_\text{P}$. SAM inherently tackles task ambiguities with a multi-mask approach that follows Bayesian concepts to estimate epistemic uncertainty. SAM generates three masks representing predictions for different $\hat{a} \in \hat{\mathcal{A}}$ using different random input embeddings (see Figure 2). However, the true task probability $p(\hat{a}|x_\text{I}, x_\text{P}, \theta)$ is intractable.

Reviewing the former mentioned methods, MC variations of $\theta$ like ensembles or BNNs to estimate epistemic uncertainty are not applicable post-hoc and are highly cost-intensive for foundation models. Aleatoric approximations that augment the prompt $x_\text{P}$ (Deng et al., 2023; Zhang et al., 2023b) only address aleatoric uncertainty that stems from memorized biases in the task decoder. Image augmentations like in (Wang et al., 2019; Lahlou et al., 2023) may only address the image encoder, neglecting prompt-induced uncertainties. It shows that the task-agnostic nature of SAM challenges current UQ strategies. To address these challenges, we propose a more fine-grained decomposition of uncertainty in SAM. Beyond epistemic uncertainty, we subdivide aleatoric uncertainty into prompt and task uncertainty.[2]

## 3. Method

As shown, SAM necessitates a more fine-grained evaluation of uncertainty. First, we formulate Bayesian approximations of the predictive, epistemic, prompt, and task uncertainty to evaluate known ideas and to baseline our main method. To make UQ usable for applications, we secondly present USAM that directly estimates the sources of uncertainty from SAM's latent without sampling or retraining.

The task is to quantify the probability that SAM's fore-

---

[2]During the conference, *Kirchhof et al.* initiated a discussion on new perspectives regarding uncertainty in large language models, highlighting limitations in the commonly made distinction between aleatoric and epistemic uncertainty (Kirchhof et al., 2025). These concepts are also applicable to this paper, particularly their term *"Underspecification"* that highly relates to Equation (2).

ground mask estimation $y$ is correct, *i.e.* is equal to the ground truth $\overline{y} = a$. The uncertainty can be quantified for a single pixel $y$ or entire masks $\mathbf{y}$. This paper quantifies the latter to address applications that utilize per image evaluations like supervision management (Vujasinović et al., 2024) or adaptive model scaling (Aggarwal et al., 2024).

### 3.1. A Bayesian Entropy Approximation

The predictive uncertainty from Equation (2) is intractable. To address this, we approximate it using ideas from existing sampling-based methods (Wang et al., 2019; Deng et al., 2023; Lakshminarayanan et al., 2017). Specifically, we predict a set of two-dimensional foreground probability maps $\mathcal{Y} = \{\mathbf{y}^1, \ldots, \mathbf{y}^K\}$ using SAM, where the cardinality is defined as $K = |\mathcal{T}| \cdot |\mathcal{X}_\text{P}| \cdot |\hat{\mathcal{A}}| \cdot |\Theta|$. It combines the sets of image augmentations $t \in \mathcal{T}$ (aleatoric), randomly sampled point-coordinate prompts $x_\text{P} \in \mathcal{X}_\text{P}(a)$ drawn equally distributed from the ground-truth mask (prompt), the tasks $\hat{\mathcal{A}}$ encoded in SAMs mask proposals (task), and all available pre-trained models $\Theta$ of different size (epistemic). The models are denoted as $\Theta = \{\text{L}, \text{B+}, \text{S}, \text{T}\}$ and correspond to the names *Large*, *Base+*, *Small*, and *Tiny*. We compute the task probability $p(\hat{a}|t(x_\text{I}), x_\text{P}, \theta)$ by normalizing over all *SamScores* of a prediction. Using the weighted entropy

$$
H(\mathbf{y}) = -\sum_{y \in \mathbf{y}} \frac{y}{\|\mathbf{y}\|_1} \big( y \log_2(y) + (1-y) \log_2(1-y) \big) , \quad (3)
$$

the predictive uncertainty is quantified for an entire image by calculating the weighted sum of $\mathcal{Y}$

$$
H_\mathcal{Y}(x_\text{I}, a, \mathcal{T}, \Theta, \hat{\mathcal{A}}, \mathcal{Y}) \approx
$$
$$
H\left( \sum_{\substack{(t, x_\text{P}, \theta, \hat{a}) \in \\ \mathcal{T} \times \mathcal{X}_\text{P}(a) \times \Theta \times \hat{\mathcal{A}}}} \frac{p(\hat{a}|t(x_\text{I}), x_\text{P}, \theta)}{|\mathcal{T}| \cdot |\mathcal{X}_\text{P}(a)| \cdot |\Theta|} \mathbf{y}_{t(x_\text{I}), \hat{a}, \theta, x_\text{P}} \right), \quad (4)
$$

where the indices of $\mathbf{y}_\bullet \in \mathcal{Y}$ indicate the prediction configuration. Higher entropy $H_\mathcal{Y}$ indicates greater uncertainty, though the specific source of uncertainty remains unknown. Thus, we compute entropies for distinct uncertainty sources. The epistemic model uncertainty is quantified by

$$
H_\Theta(x_\text{I}, x_\text{P}, a, \Theta, \hat{\mathcal{A}}, \mathcal{Y}) \approx H\left( \sum_{\theta \in \Theta} \frac{1}{|\Theta|} \mathbf{y}_{x_\text{I}, \text{Best}(a, \hat{\mathcal{A}}), \theta, x_\text{P}} \right)
$$
$$
(5)
$$

calculated for a given image $x_\text{I}$, prompt $x_\text{P}$, and the known ground truth task $a$. The operator $\text{Best}(\bullet)$ selects the best fitting mask proposal according to the ground truth and thus follows the standard SAM evaluation protocol. Similar, the prompt uncertainty is quantified by

$$
H_{\mathcal{X}_\text{P}}(x_\text{I}, \theta, a, \hat{\mathcal{A}}, \mathcal{Y}) \approx H\left( \sum_{x_\text{P} \in \mathcal{X}_\text{P}(a)} \frac{1}{|\mathcal{X}_\text{P}|} \mathbf{y}_{x_\text{I}, \text{Best}(a, \hat{\mathcal{A}}), \theta, x_\text{P}} \right)
$$
$$
(6)
$$

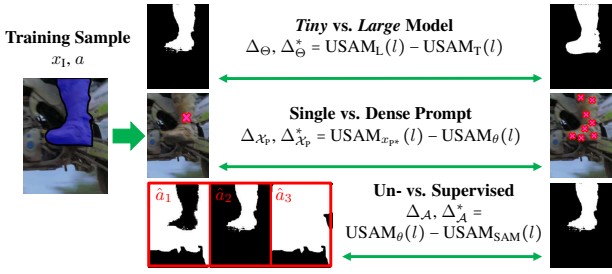

*Figure 3.* Training objectives of our MLPs. They estimate the gap between simple and cheap (left) and refined (right) predictions.

and requires pre-selected model weights. Finally, we quantify the task uncertainty with all three mask proposals:

$$H_{\mathcal{A}}(x_{\mathrm{I}}, x_{\mathrm{P}}, \theta, \hat{\mathcal{A}}, \mathcal{Y}) \approx H\left(\sum_{\hat{a} \in \hat{\mathcal{A}}} p(\hat{a}|x_{\mathrm{I}}, x_{\mathrm{P}}, \theta)\mathbf{y}_{x_{\mathrm{I}}, \hat{a}, \theta, x_{\mathrm{P}}}\right). \tag{7}$$

This Bayesian framework allows the quantification of predictive uncertainty $H_{\mathcal{Y}}$, epistemic model uncertainty $H_{\Theta}$, aleatoric prompt uncertainty $H_{\mathcal{X}_{\mathrm{P}}}$, and aleatoric task uncertainty $H_{\mathcal{A}}$ through MC simulations. They allow evaluating known concepts for the task-agnostic SAM setting.

More details to $\mathcal{T}$, $\mathcal{X}_{\mathrm{P}}$, $\hat{\mathcal{A}}$, and $\Theta$ are given in Appendix A. Note that $\hat{\mathcal{A}}$ depends on the respective $x_{\mathrm{I}}$, $x_{\mathrm{P}}$, and $\theta$ which we omit in the equations due to space limitations.

### 3.2. USAM: A Deterministic Estimator

While the Bayesian entropy approximation closely reflects Equation (2), the computational effort is huge, ground truth is required, and a user needs to draw multiple prompts. To address these limitations, we introduce USAM and extend SAM with lightweight estimators that directly estimate uncertainty proxies. We do this by concatenating SAMs 256-dimensional mask and IoU tokens, further denoted as $l$, and feeding them into MLPs that are trained to predict uncertainty as indicated in Figure 2. The simple design of the MLPs aligns with the architecture of SAM's inherent MLP for predicting the SamScore, as described in the Appendix of (Kirillov et al., 2023). They have three layers and a sigmoid activation, but we use 512 input dimensions and hidden states due to the concatenated input tokens. All MLPs are trained to minimize the mean squared error to the objectives that are discussed next and visualized in Figure 3.

For each dataset image $x_{\mathrm{I}} \in \mathcal{X}$, we predict the set $\mathcal{Y}$ and extract the tokens $l$. Additionally, we predict $y_{\mathrm{P}}^{*}$ using a refined prompt $x_{\mathrm{P}}^{*} \cup \mathcal{X}_{\mathrm{P}}$ that consists of all single-coordinate prompts and represents a user that spends high-effort to create a prompt with high certainty. We calculate the IoU values from $\mathcal{Y}$ to the ground truth to train USAM. Given an arbitrary $l$ that can stem from any model in $\Theta$, our first MLP $\mathrm{USAM}_{x_{\mathrm{P}*}}(l)$ estimates the IoU that is achieved us-

ing the respective input image $x_{\mathrm{I}}$ and the refined prompt $x_{\mathrm{P}}^{*}$. Furthermore, $\mathrm{USAM}_{\mathrm{T}}(l)$, $\mathrm{USAM}_{\mathrm{S}}(l)$, $\mathrm{USAM}_{\mathrm{B+}}(l)$, and $\mathrm{USAM}_{\mathrm{L}}(l)$ estimate the expected IoU that the *Tiny*, *Small*, *Base+*, and *Large* SAM model achieve, respectively, and are in summary denoted as $\mathrm{USAM}_{\theta}(l)$. The last MLP $\mathrm{USAM}_{\mathrm{SAM}}(l)$ estimates the accuracy achieved with the mask selected by the *SamScore*, *i.e.* without applying $\mathrm{Best}(\bullet)$ and comparing to the ground truth as done in SAMs evaluation.

During test-time with a given SAM model $\theta$ that creates the tokens $l$, we quantify the predictive uncertainty by predicting and inverting the expected IoU with $1 - \mathrm{USAM}_{\theta}(l)$. A low expected IoU indicates a higher uncertainty without indicating the cause. To quantify the prompt uncertainty, we calculate the expected gap $\Delta_{\mathcal{X}_{\mathrm{P}}}(l) = \mathrm{USAM}_{x_{\mathrm{P}*}}(l) - \mathrm{USAM}_{\theta}(l)$ between the single-coordinate and refined prompt. Similar, we quantify the task uncertainty by estimating the gap $\Delta_{\mathcal{A}}(l) = \mathrm{USAM}_{\theta}(l) - \mathrm{USAM}_{\mathrm{SAM}}(l)$ between the supervised and unsupervised mask selection. The model uncertainty is quantified with the expected gap $\Delta_{\Theta}(l) = \mathrm{USAM}_{\mathrm{L}}(l) - \mathrm{USAM}_{\mathrm{T}}(l)$ between *Large* and *Tiny* SAM. Since all models are trained equally and only differ in size, we expect high epistemic uncertainties causing the largest drop in the tiniest model.

Furthermore, additional MLPs estimate $\Delta_{\bullet}(l)$ directly from the latent and are denoted as $\Delta_{\bullet}^{*}(l)$. They aim to recognize patterns that indicate specific uncertainties. The direct MLPs estimate the expected gap without the expected IoU, thus focusing on the relevant aspects during optimization while further reducing computation. USAM is a lightweight alternative to the Bayesian approximation.

## 4. Experiments

This paper presents a theoretical close approximation and the efficient USAM to quantify SAM's uncertainty. The following section presents experimental results that investigate the methods and highlight the effectiveness for practical tasks. First, we describe the experimental settings. Then, we analyse the ability to estimate the uncertainty that lies in insufficient models, inaccurate prompts, or ambiguous tasks. Finally, we analyze the overall segmentation uncertainty and present qualitative results of our MLPs.

### 4.1. Setting

In our experiments, we use SAM models pretrained by (Ravi et al., 2024) to implement the Bayesian approximation and USAM. The MLPs of USAM are trained with the SGD optimizer (weight decay 0.001). Hyperparameters are optimized using the Bayesian optimization framework SMAC3 (Lindauer et al., 2022). The number of epochs is limited to between 5 and 80, the batch size between 16 and 256, the

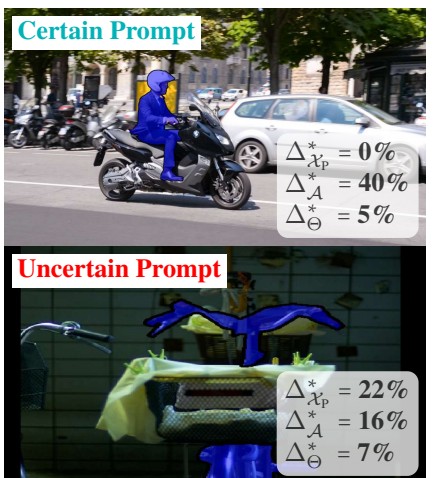
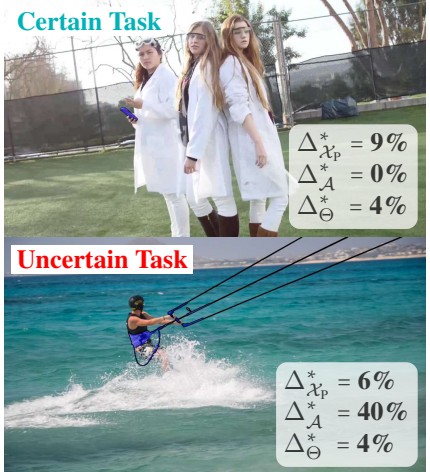
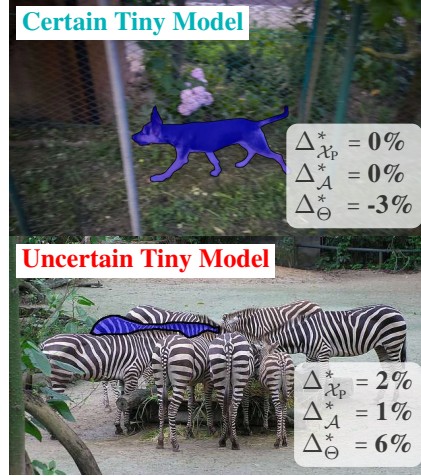

*Figure 4.* Samples with low (certain) and high (uncertain) uncertainty scores. The uncertainty is quantified with USAM, *i.e.* the expected difference between simple and refined prompts ($\Delta_{\mathcal{X}_{\mathrm{P}}}^*$), supervised and *SamScore* selection ($\Delta_{\mathcal{A}}^*$), and the *Tiny* and *Large* model ($\Delta_{\Theta}^*$).

learning rate between 0.0001 and 0.1, and SGDs momentum between 0.1 and 0.9. During SMAC3 optimization, we split the training set into 80% training and 20% validation subsets. Our best trained models on the large-scale dataset SA-V are publicly available in our code repository.

Combining the dataset selection of (Kirillov et al., 2023) and (Ravi et al., 2024), we run experiments on different scaled instance segmentation datasets DAVIS (Perazzi et al., 2016), ADE20k (Zhou et al., 2019), MOSE (Ding et al., 2023), COCO (Lin et al., 2015), and SA-V (Ravi et al., 2024). The training is performed only using the training data of the respective dataset without access to validation data. Due to missing public test data for DAVIS, COCO, and ADE20k, we use the validation set for extensive evaluation. For similar reasons, we use the SA-V trained models to evaluate on the MOSE train data.

To rank competing methods, we employ the mean intersection over union (*a.k.a.* mIoU) of predicted and ground truth masks. The uncertainty estimation capability is evaluated with the area under curve (AUC) of the mIoU, when a variable ratio between 0% and 100% of the most uncertain samples are corrected or refined with a better estimate. Depending on the evaluated task, we correct uncertain samples with the ground truth or refine the prediction using a larger model, better prompts, or task supervison. Good uncertainty estimation methods assign large scores to samples that are likely wrong. Therefore, a correction of those samples leads to better mIoUs and the AUC increases, such that a better UQ estimation is indicated. For readability, we normalize the AUC between the best and worst possible results (rel. AUC). All corresponding absolute values can be found in the Appendix. We follow (Ravi et al., 2024; Kirillov et al., 2023) and choose the best mask proposal during evaluation.

*Table 1.* Model uncertainty quantification. The table shows the area under curve (AUC) when predicting a variable fraction of the most uncertain samples with the *Large* model, others with the tiny one. The uncertainty is determined by the respective method.

| [rel. AUC in %]↑ | | DAVIS | ADE20k | MOSE | COCO | SA-V |
|---|---|---|---|---|---|---|
| **SAM** | *SamScore* | 64.25 | 52.85 | 68.44 | 57.97 | 58.83 |
| | Entropy | 71.78 | 53.48 | 71.56 | 61.61 | 63.49 |
| **Bayes** | $H_{\mathcal{Y}}$ | 51.48 | 52.06 | 61.36 | 54.27 | 55.39 |
| | $H_{\Theta}$ | **73.46** | 57.65 | **75.60** | 64.43 | **65.66** |
| **USAM** | $\Delta_{\Theta}$ | 66.85 | 60.32 | 71.78 | 63.66 | 61.23 |
| | $\Delta_{\Theta}^*$ | 59.08 | **61.55** | 73.05 | **66.60** | 63.71 |

We compare our MLPs to the introduced Bayesian baseline and to further standard UQ approaches derived from SAM. On one hand, we use the inverse *SamScore* as an indicator for uncertainty. A larger *SamScore* indicates a higher expected IoU and is a proxy for UQ. Furthermore, we use the mean entropy of the pixel-wise foreground probabilities that correspond to the predicted mask and denote it as $H_{\mathrm{Std}}$. Mask entropy is often used as UQ method in applications, *e.g.* in (Vujasinović et al., 2024).

## 4.2. Model Uncertainty

As commonly known, larger models have a larger predictive ability compared to smaller ones. This is also true for SAM. However, even if large models are often superior, they go along with larger energy consumption and hardware requirements (Glandorf et al., 2023). Ideally, a large model should only be used when the small model is significantly more uncertain and is expected to have a lower accuracy. Thus, we evaluate the ability to estimate the uncertainty stemming from the model. We predict the test set with the *Tiny* model

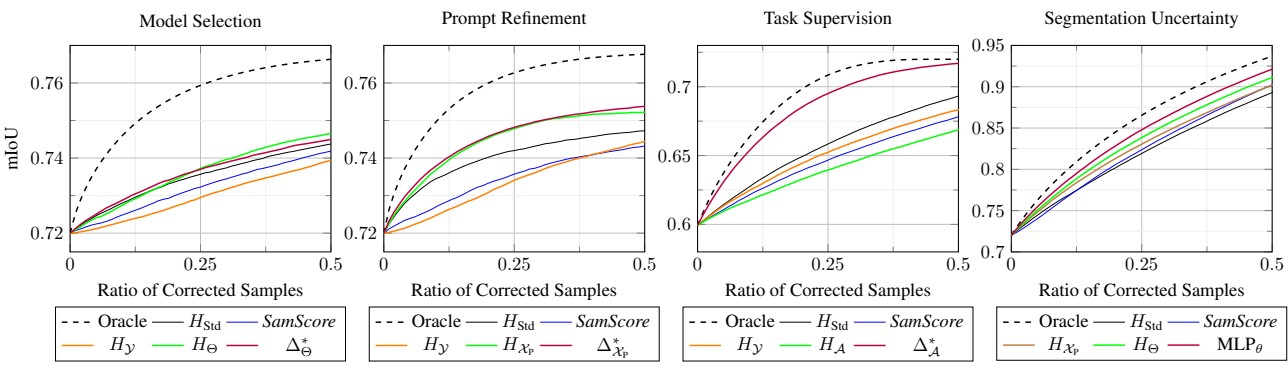

*Figure 5.* Performance gain while improving predictions selected with UQ on the COCO dataset. We evaluate the *SamScore*, mask entropy $H_{\text{Std}}$, the Bayesian entropy approximations $H_{\mathcal{Y}}$, $H_{\mathcal{A}}$, $H_{\mathcal{X}_{\text{P}}}$, $H_{\Theta}$ and our USAM$_\theta$, $\Delta^*_{\mathcal{A}}$, $\Delta^*_{\mathcal{X}_{\text{P}}}$, $\Delta^*_{\Theta}$. The dashed line denotes an oracle estimation. Beginning from left to right, the first plot shows the improvement when replacing a ratio of the most uncertain predictions of the *Tiny* model with the *Large* models. The second plot shows the improvement, when using refined prompts to the most uncertain samples. The third and foruth, when the best mask in $\hat{\mathcal{A}}$ is selected ignoring the *SamScore* and if a ratio of the most uncertain predictions is replaced by the ground truth. It shows that our MLPs are consistently superior to all other methods or on-par with the Bayesian approximation.

*Table 2.* Prompt uncertainty quantification. The table shows the area under curve (AUC) when predicting a variable fraction of the most uncertain samples with a refined prompt containing multiple point coordinates, others with a single-coordinate prompt. The uncertainty is determined by the respective method.

| [rel. AUC in %] ↑ | | DAVIS | ADE20k | MOSE | COCO | SA-V |
|---|---|---|---|---|---|---|
| SAM | *SamScore* | 71.82 | 69.12 | 66.85 | 60.55 | 54.13 |
| | Entropy | 77.13 | 70.15 | 69.24 | 68.05 | 63.56 |
| Bayes | $H_{\mathcal{Y}}$ | 54.11 | 60.82 | 63.74 | 61.25 | 55.78 |
| | $H_{\mathcal{X}_{\text{P}}}$ | **80.75** | 79.41 | 74.33 | 74.20 | 67.69 |
| USAM | $\Delta_{\mathcal{X}_{\text{P}}}$ | 75.04 | 83.23 | 74.21 | 78.17 | 70.15 |
| | $\Delta^*_{\mathcal{X}_{\text{P}}}$ | 75.53 | **83.41** | **74.50** | **78.89** | **71.27** |

*Table 3.* Task uncertainty quantification. The table shows the area under curve (AUC) when predicting a variable fraction of the most uncertain samples with the correct task, otherwise with the one selected by the SamScore. The most uncertain samples are determined by the respective method.

| [rel. AUC in %] ↑ | | DAVIS | ADE20k | MOSE | COCO | SA-V |
|---|---|---|---|---|---|---|
| SAM | *SamScore* | 68.36 | 66.31 | 70.58 | 64.09 | 58.87 |
| | Entropy | 68.77 | 76.49 | 73.55 | 74.56 | 70.88 |
| Bayes | $H_{\mathcal{Y}}$ | 74.88 | 64.53 | 67.32 | 68.12 | 70.50 |
| | $H_{\mathcal{A}}$ | 43.86 | 52.79 | 66.04 | 57.55 | 78.13 |
| USAM | $\Delta_{\mathcal{A}}$ | 94.05 | **93.08** | **94.21** | 94.85 | 94.17 |
| | $\Delta^*_{\mathcal{A}}$ | **94.31** | 92.38 | 94.01 | **94.87** | **94.61** |

and re-predict a ratio of the most uncertain samples with the *Large* model. The ratio parameter reflects the trade-off between accuracy and energy consumption.

Table 1 shows the AUC of the mIoU with a corresponding plot given in Figure 5 (*Model Selection*). The Bayesian entropy $H_{\Theta}$ and our direct USAM MLP $\Delta^*_{\Theta}$ are on-par and the most precise UQ methods. Moreover, the default UQ method *SamScore* is consistently outperformed by all other methods, except from $H_{\mathcal{Y}}$. The weak performance of $H_{\mathcal{Y}}$ is reasonable, because it includes all uncertainty aspects including the task and prompt uncertainty that are irrelevant and potentially misleading for this evaluation. However, USAM performs often best and requires negligible computational efforts compared to the Bayesian entropy and is therefore the preferred method to preserve energy. The only setting in which USAM is not competitive is on the small-scale DAVIS dataset, indicating that more data variance is required for training which is also observable in the next experiments. Additional insights about the impact of the model size are briefly analyzed in the Appendix: Ta-

ble 9 shows that SAM with the largest backbone is superior in most situations on all datasets. Furthermore, Table 10 shows that the large model is notably more robust to image corruptions and noise.

### 4.3. Prompt Uncertainty

Similar to the former experiment, we evaluate the ability to quantify uncertainty stemming from prompts. We do this by predicting the dataset with a simple coordinate prompt in the centroid of the mask and replace a ratio of the most uncertain prompts with refined ones consisting of 8 equally distributed coordinate prompts from the ground truth mask.

The AUC of the mIoU is reported in Table 2 with a corresponding plot in Figure 5 (*Prompt Refinement*). Similar to the former experiment, the Bayesian entropy $H_{\mathcal{X}_{\text{P}}}$ and our direct USAM MLP $\Delta^*_{\mathcal{X}_{\text{P}}}$ perform superior with $\Delta^*_{\mathcal{X}_{\text{P}}}$ being slightly better. The experiment shows that our method can be used to notify users when prompts lead to uncertain results.

*Table 4.* Uncertainty quantification. The table shows the area under curve (AUC) when correcting a variable fraction of the most uncertain samples. The most uncertain samples are determined by the respective method. The non-corrected samples are predicted with the *Large* SAM model. Similar numbers are given for the *Tiny*, *Small*, and *Base+* models in the Appendix, Table 14.

| [rel. AUC in %]↑ | | DAVIS | ADE20k | MOSE | COCO | SA-V |
|---|---|---|---|---|---|---|
| SAM | *SamScore* | 83.36 | 75.40 | 89.48 | 79.60 | 80.80 |
| | $H_{\text{Std}}$ | 82.00 | 70.23 | 82.09 | 74.27 | 79.60 |
| Bayes | $H_{\mathcal{Y}}$ | 38.79 | 46.87 | 56.58 | 45.36 | 57.32 |
| | $H_{\Theta}$ | 84.24 | 74.39 | 84.02 | 78.26 | 85.90 |
| | $H_{\mathcal{X}_{\text{P}}}$ | 87.16 | 83.79 | 86.23 | 79.13 | 83.23 |
| | $H_{\mathcal{A}}$ | 52.24 | 52.69 | 66.07 | 56.96 | 60.51 |
| | **USAM$_{\text{L}}$** | **91.59** | **92.60** | **92.78** | **90.01** | **89.37** |

*Table 5.* Pearson correlation between different UQ measures on the COCO validation dataset using by the *Large* SAM model. IoU$_{\text{GT}}$ denotes the real intersection over union between SAMs prediction and the ground truth. Equivalent tables obtained by the other SAM models are available in the Appendix, Tables 15 to 17.

| | IoU$_{\text{GT}}$ | SamS | $H_{\text{Std}}$ | $H_{\mathcal{Y}}$ | $H_{\Theta}$ | $H_{\mathcal{A}}$ | $H_{\mathcal{X}_{\text{P}}}$ | USAM$_{\text{L}}$ | $\Delta^*_{\Theta}$ | $\Delta^*_{\mathcal{A}}$ | $\Delta^*_{\mathcal{X}_{\text{P}}}$ |
|---|---|---|---|---|---|---|---|---|---|---|---|
| IoU$_{\text{GT}}$ | 1.00 | 0.49 | -0.30 | 0.10 | -0.54 | -0.09 | -0.42 | 0.71 | -0.26 | -0.17 | -0.15 |
| SamS | 0.49 | 1.00 | -0.44 | -0.09 | -0.63 | -0.14 | -0.41 | 0.58 | -0.38 | -0.53 | -0.24 |
| $H_{\text{Std}}$ | -0.30 | -0.44 | 1.00 | 0.41 | 0.81 | 0.36 | 0.74 | -0.38 | 0.69 | 0.49 | 0.66 |
| $H_{\mathcal{Y}}$ | 0.10 | -0.09 | 0.41 | 1.00 | 0.26 | 0.42 | 0.39 | 0.17 | 0.23 | 0.33 | 0.27 |
| $H_{\Theta}$ | -0.54 | -0.63 | 0.81 | 0.26 | 1.00 | 0.33 | 0.74 | -0.60 | 0.62 | 0.45 | 0.51 |
| $H_{\mathcal{A}}$ | -0.09 | -0.14 | 0.36 | 0.42 | 0.33 | 1.00 | 0.36 | -0.11 | 0.31 | 0.17 | 0.17 |
| $H_{\mathcal{X}_{\text{P}}}$ | -0.42 | -0.41 | 0.74 | 0.39 | 0.74 | 0.36 | 1.00 | -0.42 | 0.61 | 0.39 | 0.53 |
| USAM$_{\text{L}}$ | 0.71 | 0.58 | -0.38 | 0.17 | -0.60 | -0.11 | -0.42 | 1.00 | -0.37 | -0.23 | -0.21 |
| $\Delta^*_{\Theta}$ | -0.26 | -0.38 | 0.69 | 0.23 | 0.62 | 0.31 | 0.61 | -0.37 | 1.00 | 0.35 | 0.72 |
| $\Delta^*_{\mathcal{A}}$ | -0.17 | -0.53 | 0.49 | 0.33 | 0.45 | 0.17 | 0.39 | -0.23 | 0.35 | 1.00 | 0.29 |
| $\Delta^*_{\mathcal{X}_{\text{P}}}$ | -0.15 | -0.24 | 0.66 | 0.27 | 0.51 | 0.17 | 0.53 | -0.21 | 0.72 | 0.29 | 1.00 |

### 4.4. Task Uncertainty

During evaluation of the class-agnostic open set segmentation problem, the best mask proposal per sample is used for evaluation. This does not reflect the workflow in real-world pipelines, in which the mask with the highest *SamScore* is chosen. Thus, uncertainty stemming from ambiguous tasks with multiple valid results can be used to ask for supervision.

The UQ from ambiguous tasks is evaluated in Table 3. Different to the former experiments, we choose SAMs mask proposal with the highest *SamScore*. Then, for a ratio of the most uncertain samples, we replace it by the mask proposal that is best fitting to the ground truth. USAM is performing best with a large gap as visualized in Figure 5 (*Task Supervision*). Interestingly, the Bayesian entropy is not a suitable approximation in this setting. However, the quality of our USAM is superior and the favorable method.

### 4.5. Segmentation Uncertainty

Uncertainty correlates to the expected segmentation error. Thus, we measure the capability of predictive uncertainty

*Table 6.* Token ablation. The UQ performance of USAM when removing mask or IoU tokens from the MLP input on the COCO dataset, measured in relative AUC as in the main experiments.

| Mask Token | IoU Token | Model Uncertainty | Prompt Uncertainty | Task Uncertainty |
|---|---|---|---|---|
| ✗ | ✓ | 61.19% | 72.08% | 86.89% |
| ✓ | ✗ | 62.63% | 76.42% | 91.96% |
| ✓ | ✓ | **63.66%** | **78.30%** | **94.82%** |

*Table 7.* Runtime of SAM with and without UQ methods on a regular image performed on a *NVIDIA RTX3050 Ti*. Entropy is calculated on SAMs logit map, $|\mathcal{T}|$ and $|\mathcal{X}_{\text{P}}|$ denote the number of applied image and prompt augmentations used for MC sampling, and USAM contains the calculation of all our proposed MLPs. Compared to the runtime of all other UQ methods, our USAM is faster and easier to implement.

| [$\frac{\text{Seconds}}{\text{Iteration}}$]↓ | SAM | +Entropy | +$|\mathcal{T}| = 5$ | +$|\mathcal{X}_{\text{P}}| = 8$ | +USAM |
|---|---|---|---|---|---|
| Large | 0.437 | 0.452 | 2.187 | 0.500 | **0.441** |
| Base+ | 0.205 | 0.233 | 1.028 | 0.289 | **0.210** |
| Small | 0.134 | 0.157 | 0.688 | 0.232 | **0.142** |
| Tiny | 0.122 | 0.149 | 0.584 | 0.198 | **0.139** |

quantification with the mIoU when correcting a ratio of the most uncertain samples with the ground truth. Table 4 presents the AUC of the mIoU scores. Without exception, our MLP USAM$_{\text{L}}$ that predicts the expected IoU clearly outperforms all other methods. Interestingly, the prompt and model uncertainty $H_{\mathcal{X}_{\text{P}}}$ and $H_{\Theta}$ lead to acceptable results, too, indicating that those are the main causes of predictive uncertainty. It is important to note that the *SamScore* is not on-par even if it is optimized to solve exactly this task during SAM's training. A corresponding plot is given in Figure 5 (*Segmentation Uncertainty*).

### 4.6. Qualitative results and Discussion

Visual examples reveal insights about SAM's capabilities as shown in Figure 4 and in the Appendix. On the left side, samples with high and low prompt uncertainty $\Delta_{\mathcal{X}_{\text{P}}}$

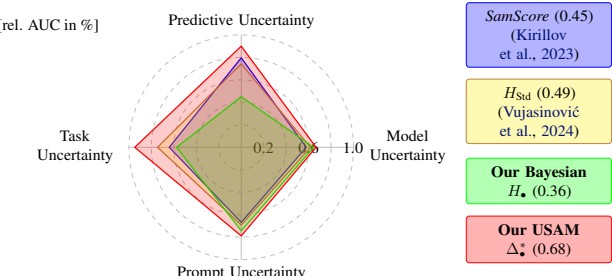

*Figure 6.* Uncertainty quantification capabilities on the COCO Dataset extracted from Tables 1 to 4 with their enclosed areas.

are shown. With a single coordinate prompt in the centroid of the blue highlighted mask, SAM is able to segment the person, but struggles with the partially occluded bike. Better prompts could help to indicate the segments. In the middle column, the upper image has high task certainty $\Delta_{\mathcal{A}}$, because a coordinate prompt on the mobile is not pointing to other objects (*e.g.* the girl). Contrary, the centroid of the blue area in the lower image could belong to the person, the kite, or both together. With respect to the model uncertainty $\Delta_\Theta$ in the right column, visual inspection shows that the *Tiny* model is consistently accurate on clearly visible dogs, but drops on homogeneous structures as the zebras compared to the *Large* model. This indicates different strengths of the pre-trained SAM models.

A view on the correlations between our calculated values in Table 5 helps to understand the behaviour of uncertainty and UQ. Compared to our MLP that estimates the IoU, the *SamScore* has a lower correlation to the overall uncertainty IoU$_{\text{GT}}$. This is interesting, because both are trained on the same objective and ours only differs in the usage of the mask token as input, the optimization with SMAC3, and is trained post-hoc. Furthermore, the IoU$_{\text{GT}}$ is highly correlated to the Bayesian model and prompt entropy ($H_\Theta$, $H_{\mathcal{X}_\text{P}}$) that are explicitly modelled in this paper. The large correlation to $H_\Theta$ supports the findings that a lot of uncertainty stems from the model (Appendix, Tables 9 and 10).

The runtime comparison presented in Table 7 shows that the MLPs of our USAM perform UQ efficiently with the lowest computational overhead. Bayesian methods based on MC sampling heavily increase the computation time depending on the number of samples. Also the calculation of the logit entropy is slower than the lightweight execution of our proposed MLPs.

To further investigate our method, we set either the IoU token or the mask token to 0 to remove potentially informative patterns. We evaluate using the tiny model across the three tasks described in the main experiments performed on the COCO dataset. The results that are shown in Table 6 indicate that both tokens contribute to the predictive capability, yet they remain accurate as standalone features. Both in combination lead to the best results.

All in all, the lightweight and straight forward USAM leads to unexpected superior results which outperforms all other methods. A graphical comparison with corresponding enclosing area values is visualized in Figure 6. Since all results we present are applied post-hoc, an integration into the cost-intensive training of SAM may lead to improved results and should be considered in the future. We already provide USAM here. The Bayesian predictive entropy $H_\mathcal{Y}$ does not lead to competitive results in the presented practical problems. However, the well-performing $H_\Theta$, $H_{\mathcal{X}_\text{P}}$, and $H_{\mathcal{A}}$ show that Equation (4) includes all aspects that are relevant

to quantify uncertainty and can be used for benchmarking.

## 5. Conclusion

This paper formalizes UQ in the context of SAM by splitting it into epistemic model as well as aleatoric prompt and task uncertainty. To model those uncertainties, we evaluate a theoretically grounded Bayesian entropy approximation. To overcome the computational costs, we introduce USAM that consists of deterministic MLPs to estimate the same uncertainties efficiently. Our experiments demonstrate compelling results, comparing the Bayesian and deterministic approaches with standard SAM UQ methods. We reliably identify the source of uncertainty in real-world applications, like prompt refinement, supervision of SAMs proposals, and adaptive model scaling. Remarkably, our lightweight USAM perform better or on-par with the Bayesian approach which are therefore the new state-of-the-art. Despite their simplicity, the improvements delivered by USAM, even for tasks similar to those handled by the *SamScore*, underscores the potential for future iterations of SAM to address UQ more effectively. Our contributions help to improve uncertainty handling mandatory in safety critical domains.

## Acknowledgments

This work was supported partially by the German Federal Ministry of the Environment, Nature Conservation, Nuclear Safety and Consumer Protection (GreenAutoML4FAS project no. 67KI32007A), the Federal Ministry of Education and Research (BMBF), Germany, under the AI service center KISSKI (grant no. 01IS22093C), the MWK of Lower Sachsony within Hybrint (VWZN4219), the Deutsche Forschungsgemeinschaft (DFG) under Germany's Excellence Strategy within the Cluster of Excellence PhoenixD (EXC2122), the European Union under grant agreement no. 101136006 – XTREME

## Impact Statement

Uncertainty quantification leads to more reliable and interpretable predictions of machine learning algorithms. Especially prior works in critical domains such as medical diagnosis show that the integration of uncertainty is important (Deng et al., 2023; Wang et al., 2019; Zhang et al., 2023b). This paper contributes to better knowledge about uncertainty in the context of the popular *Segment Anything Model* that is publicly available and employed in myriad applications. Additionally, our results have the potential to decrease the carbon footprint of those applications.

To the best of our knowledge, uncertainty quantification in general is not a topic of interest for unethical applications or dual-use research of concerns.

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

# A. Details to *A Bayesian Entropy Approximation*

Section 3 introduces a Bayesian entropy approximation that formalizes a UQ baseline for our evaluation based on known Monte Carlo sampling concepts. For better reproducability, we recapitulate some main aspects with more detailed notations.

The predictive uncertainty from Equation (2) is intractable. Thus, we approximate it by combining ideas of former sampling methods (Wang et al., 2019; Deng et al., 2023; Lakshminarayanan et al., 2017). We predict a set of predictions $\mathcal{Y} = \{y^1, \ldots, y^K\}$. The cardinality is defined by multiple sampling sets $K = |\mathcal{T}| \times |\mathcal{X}_\mathrm{P}| \times |\hat{\mathcal{A}}| \times |\Theta|$. We use sets of 5 image augmentations $t \in \mathcal{T}$, the identity (*i.e.* no pertubation), vertical flip, JPEG compression (10% and 30% image quality), Gaussian blur with a kernel size of 5 pixel, and Gaussian noise. The prompts $x_\mathrm{P} \in \mathcal{X}_\mathrm{P}$ are 8 randomly sampled point-coordinate drawn equally distributed from the ground-truth mask that corresponds to $a$. The 3 task estimations $\hat{\mathcal{A}}$ are defined by SAMs mask predictions and encode the three most likely tasks. Finally, the 4 publicly available pre-trained models of different size define the set of weights $\Theta = \{\mathrm{L}, \mathrm{B+}, \mathrm{S}, \mathrm{T}\}$ (*Large*, *Base+*, *Small*, and *Tiny*). It is important to note that the prompts $\mathcal{X}_\mathrm{P}$ are depending on $x_\mathrm{I}$ and $a$, and the estimated mask proposals $\hat{\mathcal{A}}$ depend on the image $x_\mathrm{I}$, prompt $x_\mathrm{P}$, and model $\theta$. Correctly, they need to be denoted as $\mathcal{X}_\mathrm{P}(x_\mathrm{I}, a)$ and $\hat{\mathcal{A}}(x_\mathrm{I}, x_\mathrm{P}, \theta)$. We omit the correct notation due to space limitations. We assume that the relation is evident and self-explaining in the equations.

The probabilities in Equation (2) underlined with *Prompt Decoder*, *Prompt Encoder*, and *Epistemic* are not defined. Thus, we set the prompt probability and the model probability to

$$p\big(x_\mathrm{P}|t(x_\mathrm{I}), a\big) = \frac{1}{|\mathcal{X}_\mathrm{P}|} \quad \text{and} \tag{8}$$

$$p\big(\theta|\mathcal{D}\big) = \frac{1}{|\Theta|}. \tag{9}$$

Moreover, we approximate the task probability $p\big(\hat{a}|t(x_\mathrm{I}), x_\mathrm{P}, \theta\big)$ using the SamScore. Since SAM was trained on a large scale dataset with a large set of tasks, we assume SAM to inherently estimate a better IoU to more frequent, *i.e.* more likely, tasks. We normalize over all three SamScores of a prediction to get the task probability:

$$p(\hat{a}|t(x_\mathrm{I}), x_\mathrm{P}, \theta) = \frac{\mathrm{SamScore}(\hat{a})}{\sum_{a \in \hat{\mathcal{A}}_{t(x_\mathrm{I}), x_\mathrm{P}, \theta}} \mathrm{SamScore}(a)} \tag{10}$$

Applying the sampling sets and the probability definitions, the predictive uncertainty is defined as

$$p(y = 1|x, \mathcal{D}, a) \approx \sum_{\substack{(t, x_\mathrm{P}, \theta, \hat{a}) \in \\ \mathcal{T} \times \mathcal{X}_\mathrm{P} \times \Theta \times \hat{\mathcal{A}}}} \frac{p\big(\hat{a}|t(x_\mathrm{I}), x_\mathrm{P}, \theta\big)}{|\mathcal{T}| \cdot |\mathcal{X}_\mathrm{P}| \cdot |\Theta|} p\big(y|t(x_\mathrm{I}), \hat{a}, \theta\big). \tag{11}$$

All further details can be found in the main paper.

*Table 8.* Hyperparameter optimization results using (Lindauer et al., 2022) in descending order. The results are obtained during training of the USAM$_T(l)$ model trained on ADE20k. Variations are performed on the batch size, epochs, learning rate, and momentum.

| Rank | Loss (MSE) | Batch Size | Epochs | Learning Rate | Momentum | Rank | Loss (MSE) | Batch Size | Epochs | Learning Rate | Momentum |
|---|---|---|---|---|---|---|---|---|---|---|---|
| 1 | 0.02436 | 106 | 79 | 0.00131 | 0.83033 | 36 | 0.02505 | 164 | 41 | 0.00186 | 0.82966 |
| 2 | 0.02436 | 110 | 79 | 0.00128 | 0.83276 | 37 | 0.02518 | 87 | 49 | 0.01451 | 0.51234 |
| 3 | 0.02437 | 98 | 79 | 0.00118 | 0.82986 | 38 | 0.02521 | 194 | 68 | 0.0013 | 0.80417 |
| 4 | 0.02437 | 98 | 79 | 0.00116 | 0.83177 | 39 | 0.02521 | 193 | 68 | 0.00126 | 0.815 |
| 5 | 0.02438 | 105 | 79 | 0.00112 | 0.832 | 40 | 0.02532 | 224 | 68 | 0.00148 | 0.79977 |
| 6 | 0.02439 | 193 | 70 | 0.00242 | 0.81485 | 41 | 0.02536 | 208 | 75 | 0.04111 | 0.24591 |
| 7 | 0.02439 | 176 | 79 | 0.00148 | 0.82823 | 42 | 0.02555 | 27 | 72 | 0.00241 | 0.86083 |
| 8 | 0.02439 | 98 | 80 | 0.00134 | 0.83175 | 43 | 0.0256 | 129 | 72 | 0.00064 | 0.8245 |
| 9 | 0.0244 | 106 | 76 | 0.00131 | 0.83136 | 44 | 0.02561 | 19 | 75 | 0.00211 | 0.83676 |
| 10 | 0.02441 | 98 | 74 | 0.00128 | 0.83464 | 45 | 0.02564 | 93 | 68 | 0.0275 | 0.58042 |
| 11 | 0.02442 | 98 | 75 | 0.00128 | 0.83464 | 46 | 0.02566 | 199 | 30 | 0.03576 | 0.64499 |
| 12 | 0.02443 | 102 | 80 | 0.00108 | 0.82989 | 47 | 0.02569 | 80 | 61 | 0.01084 | 0.78907 |
| 13 | 0.02444 | 102 | 79 | 0.00118 | 0.83004 | 48 | 0.0257 | 215 | 40 | 0.057 | 0.64422 |
| 14 | 0.02445 | 153 | 68 | 0.00248 | 0.82004 | 49 | 0.02575 | 46 | 31 | 0.02255 | 0.25254 |
| 15 | 0.02446 | 167 | 79 | 0.00132 | 0.82778 | 50 | 0.02576 | 256 | 55 | 0.08722 | 0.52532 |
| 16 | 0.02446 | 95 | 74 | 0.00143 | 0.82605 | 51 | 0.02577 | 158 | 55 | 0.02941 | 0.70937 |
| 17 | 0.02447 | 152 | 68 | 0.00235 | 0.81844 | 52 | 0.02579 | 65 | 66 | 0.0467 | 0.17168 |
| 18 | 0.02447 | 150 | 70 | 0.00227 | 0.81485 | 53 | 0.02579 | 22 | 73 | 0.00306 | 0.83178 |
| 19 | 0.02448 | 153 | 70 | 0.0024 | 0.82183 | 54 | 0.02604 | 128 | 79 | 0.06786 | 0.48176 |
| 20 | 0.02451 | 164 | 62 | 0.00183 | 0.83002 | 55 | 0.02624 | 220 | 41 | 0.03376 | 0.86587 |
| 21 | 0.02452 | 92 | 67 | 0.00128 | 0.78672 | 56 | 0.02636 | 170 | 78 | 0.04239 | 0.62245 |
| 22 | 0.02453 | 107 | 80 | 0.00215 | 0.83304 | 57 | 0.0265 | 103 | 44 | 0.0888 | 0.38777 |
| 23 | 0.02457 | 106 | 79 | 0.00208 | 0.83412 | 58 | 0.02667 | 223 | 33 | 0.03961 | 0.88511 |
| 24 | 0.02457 | 94 | 75 | 0.00214 | 0.82338 | 59 | 0.02675 | 151 | 63 | 0.07027 | 0.30754 |
| 25 | 0.02457 | 95 | 69 | 0.00227 | 0.81132 | 60 | 0.02707 | 80 | 53 | 0.06813 | 0.56828 |
| 26 | 0.02465 | 182 | 70 | 0.00144 | 0.83128 | 61 | 0.0274 | 100 | 77 | 0.00023 | 0.84102 |
| 27 | 0.0247 | 164 | 51 | 0.00198 | 0.82948 | 62 | 0.02757 | 119 | 35 | 0.03685 | 0.85836 |
| 28 | 0.02472 | 97 | 67 | 0.00245 | 0.8231 | 63 | 0.02767 | 121 | 39 | 0.03343 | 0.88359 |
| 29 | 0.02474 | 91 | 75 | 0.00301 | 0.83894 | 64 | 0.02852 | 58 | 49 | 0.05615 | 0.57001 |
| 30 | 0.02481 | 233 | 47 | 0.01569 | 0.12675 | 65 | 0.02874 | 228 | 66 | 0.00175 | 0.27064 |
| 31 | 0.02489 | 94 | 67 | 0.00105 | 0.74238 | 66 | 0.02907 | 170 | 35 | 0.09892 | 0.41356 |
| 32 | 0.02494 | 27 | 80 | 0.00131 | 0.83092 | 67 | 0.0292 | 92 | 33 | 0.00125 | 0.27468 |
| 33 | 0.02496 | 245 | 35 | 0.01568 | 0.48597 | 68 | 0.02988 | 223 | 38 | 0.0018 | 0.34614 |
| 34 | 0.025 | 164 | 36 | 0.00238 | 0.83041 | 69 | 0.02999 | 224 | 49 | 0.00143 | 0.33011 |
| 35 | 0.025 | 24 | 80 | 0.00131 | 0.83033 | 70 | 0.03083 | 23 | 71 | 0.07553 | 0.65109 |

*Table 9.* The ratio of samples per dataset where models of different size perform best. Larger models consistently perform better on average, but not on every individual sample. The performance gap between large and small models depends on the dataset.

| Ratio | | Dataset | | | |
|---|---|---|---|---|---|
| | | DAVIS | ADE20k | MOSE | COCO | SA-V |
| | Tiny | 6.38% | 18.24% | 7.85% | 15.64% | 15.72% |
| Model Size | Small | 10.40% | 20.55% | 11.06% | 17.65% | 18.39% |
| | Base+ | 27.76% | 27.69% | 25.63% | 27.46% | 28.01% |
| | Large | 55.44% | 33.50% | 55.44% | 39.24% | 37.86% |

*Table 10.* The impact of image augmentations and noise (Averaged over all datasets) on model performances evaluated with models with different size. We flipped and scaled the image, removed color (Gray) and high-frequent patterns (JPEG), and added Gaussian blur or noise. All models are scale and flip invariant but decrease the performance when adding noise. Interestingly, the smaller model with fewer weights is more prone for most image noise.

| [mIoU in %] | | Augmentation | | | | | | |
|---|---|---|---|---|---|---|---|---|
| | | Normal | Flip | Scale | Gray | GBlur | GNoise | JPEG |
| | Tiny | 76.47 | -0.01 | -0.07 | -2.06 | -8.08 | -19.36 | -16.80 |
| Model Size | Small | 77.42 | -0.03 | -0.09 | -1.91 | -8.54 | -17.55 | -15.97 |
| | Base+ | 78.64 | -0.05 | -0.06 | -1.58 | -8.77 | -17.94 | -16.62 |
| | Large | 79.58 | -0.05 | -0.10 | -1.45 | -8.67 | -14.08 | -12.99 |

*Table 11.* Model uncertainty quantification. The table shows the area under curve (AUC) when predicting a variable fraction of the most uncertain samples with the largest model, others with the tiny one. The uncertainty is determined by the respective method. Supplementary data to Table 1. Compared to Table 1, this table shows absolute AUC. To help interpreting the values, the best (*Oracle*) and worst (*Worst*) possible correction order based on the uncertainty score is added.

| | Standard Measures | | | | |
|---|---|---|---|---|---|
| [AUC in %] | DAVIS | ADE20k | MOSE | COCO | SA-V |
| Oracle | 82.518 | 64.603 | 82.131 | 76.032 | 81.805 |
| Worst | 77.778 | 59.761 | 77.685 | 71.169 | 77.235 |
| **SAM** SamScore | 80.823 | 62.320 | 80.728 | 73.988 | 79.924 |
| $H_{\mathrm{Std}}$ | 81.180 | 62.351 | 80.867 | 74.165 | 80.137 |
| **Bayes** $H_{\mathcal{Y}}$ | 80.218 | 62.282 | 80.414 | 73.808 | 79.767 |
| $H_{\Theta}$ | **81.260** | **62.553** | **81.047** | 74.302 | **80.236** |
| $H_{\mathcal{A}}$ | 80.475 | 62.233 | 80.478 | 73.794 | 79.805 |
| $H_{\mathcal{X}_{\mathrm{p}}}$ | 81.236 | 62.424 | 81.016 | **74.320** | 80.224 |

| | Classifiers Trained on DAVIS | | | | |
|---|---|---|---|---|---|
| [AUC in %] | DAVIS | ADE20k | MOSE | COCO | SA-V |
| **USAM** $\Delta_{\Theta}$ | 80.946 | 62.396 | 80.513 | 73.852 | 79.528 |
| $\Delta_{\Theta}^{*}$ | 80.578 | 62.397 | 80.427 | 73.815 | 79.708 |

| | Classifiers Trained on ADE20k | | | | |
|---|---|---|---|---|---|
| [AUC in %] | DAVIS | ADE20k | MOSE | COCO | SA-V |
| **USAM** $\Delta_{\Theta}$ | 80.996 | 62.682 | 80.690 | 73.983 | 79.917 |
| $\Delta_{\Theta}^{*}$ | 80.417 | **62.741** | 80.326 | 73.849 | 79.834 |

| | Classifiers Trained on COCO | | | | |
|---|---|---|---|---|---|
| [AUC in %] | DAVIS | ADE20k | MOSE | COCO | SA-V |
| **USAM** $\Delta_{\Theta}$ | 81.044 | 62.451 | 80.877 | 74.265 | 80.105 |
| $\Delta_{\Theta}^{*}$ | 81.105 | 62.482 | 80.933 | **74.408** | 80.145 |

| | Classifiers Trained on SA-V | | | | |
|---|---|---|---|---|---|
| [AUC in %] | DAVIS | ADE20k | MOSE | COCO | SA-V |
| **USAM** $\Delta_{\Theta}$ | 81.038 | 62.457 | 80.856 | 74.130 | 80.034 |
| $\Delta_{\Theta}^{*}$ | 81.040 | 62.417 | 80.898 | 74.239 | 80.147 |

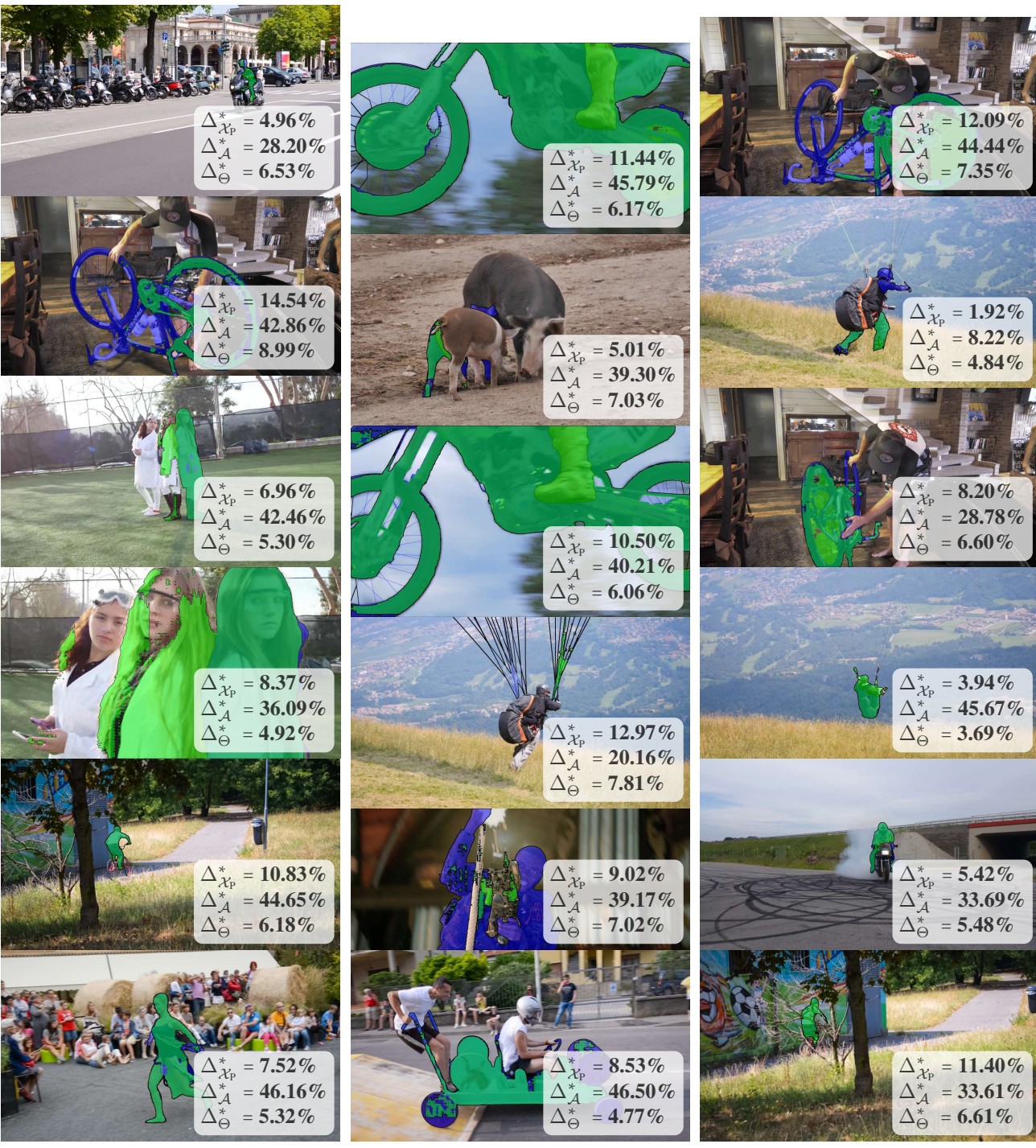

*Figure 7.* Samples with uncertainty estimations. Complementary samples to Figure 4. The ground truth mask is visualized by the blue mask and SAM's prediction of the *Tiny* model in green.

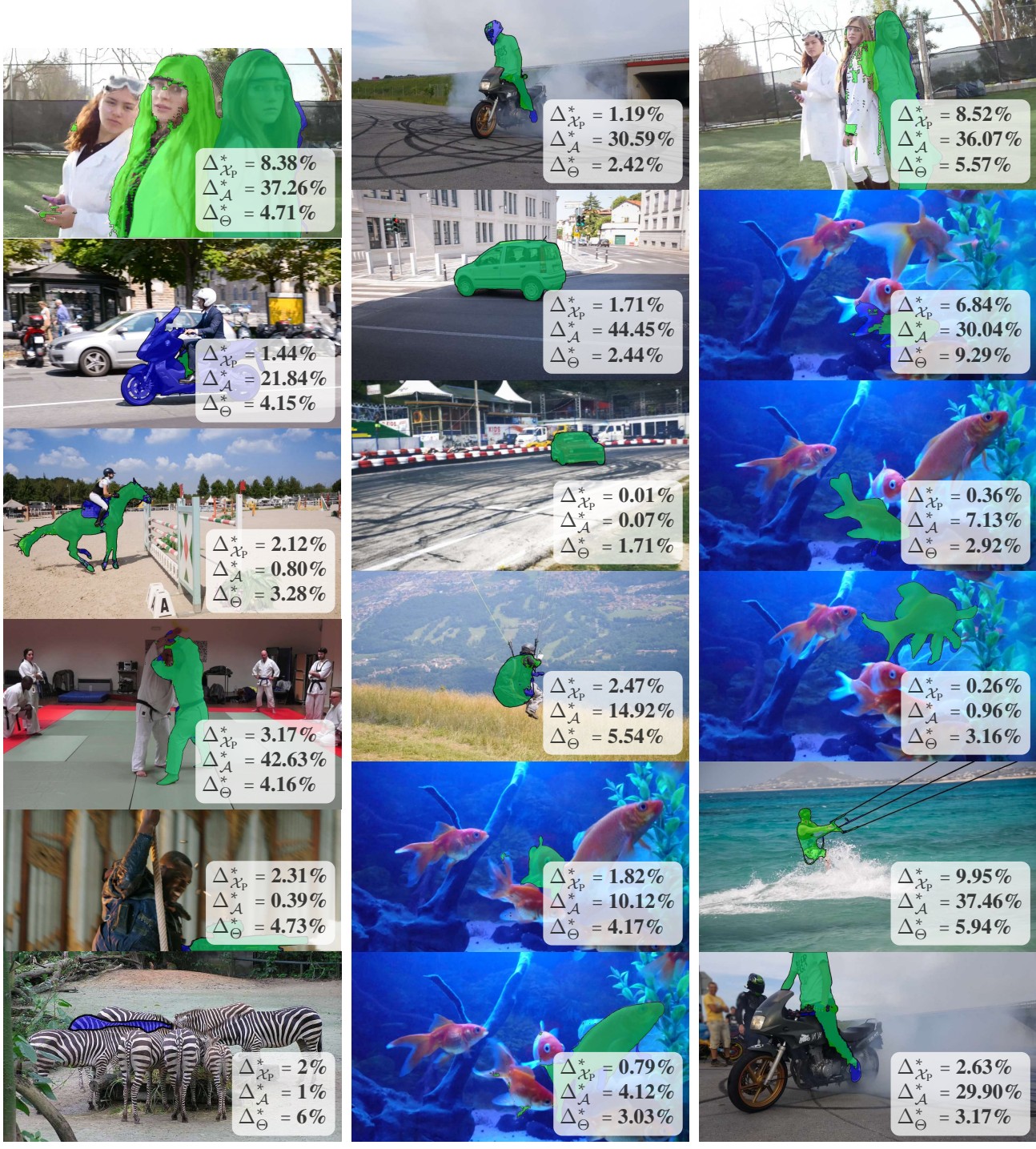

*Figure 8.* Samples with uncertainty estimations. Complementary samples to Figure 4. The ground truth mask is visualized by the blue mask and SAM's prediction of the *Tiny* model in green.

*Table 12.* Prompt uncertainty quantification. The table shows the area under curve (AUC) when predicting a variable fraction of the most uncertain samples with a refined prompt containing multiple point coordinates, others with a single-coordinate prompt. The uncertainty is determined by the respective method. Supplementary data to Table 2. Compared to Table 2, this table shows absolute AUC. To help interpreting the values, the best (*Oracle*) and worst (*Worst*) possible correction order based on the uncertainty score is added.

**Standard Measures**

| | [AUC in %] | DAVIS | ADE20k | MOSE | COCO | SA-V |
|---|---|---|---|---|---|---|
| | Oracle | 82.483 | 70.917 | 80.912 | 76.159 | 81.177 |
| | Worst | 77.500 | 61.976 | 76.523 | 70.854 | 76.341 |
| SAM | SamScore | 81.079 | 68.156 | 79.457 | 74.066 | 78.959 |
| SAM | $H_{\mathrm{Std}}$ | 81.343 | 68.248 | 79.562 | 74.464 | 79.415 |
| Bayes | $H_{\mathcal{Y}}$ | 80.197 | 67.414 | 79.321 | 74.103 | 79.039 |
| Bayes | $H_{\Theta}$ | **81.618** | 68.616 | **79.790** | 74.616 | 79.532 |
| Bayes | $H_{\mathcal{A}}$ | 80.428 | 66.723 | 79.248 | 73.794 | 78.960 |
| Bayes | $H_{\mathcal{X}_{\mathrm{P}}}$ | 81.524 | **69.077** | 79.786 | **74.790** | **79.615** |

**Classifiers Trained on DAVIS**

| | [AUC in %] | DAVIS | ADE20k | MOSE | COCO | SA-V |
|---|---|---|---|---|---|---|
| USAM | $\Delta_{\mathcal{X}_{\mathrm{P}}}$ | 81.240 | 67.801 | 79.362 | 73.931 | 79.047 |
| USAM | $\Delta^*_{\mathcal{X}_{\mathrm{P}}}$ | 81.264 | 67.671 | 79.374 | 74.084 | 78.989 |

**Classifiers Trained on ADE20k**

| | [AUC in %] | DAVIS | ADE20k | MOSE | COCO | SA-V |
|---|---|---|---|---|---|---|
| USAM | $\Delta_{\mathcal{X}_{\mathrm{P}}}$ | 81.386 | 69.418 | 79.555 | 74.563 | 79.276 |
| USAM | $\Delta^*_{\mathcal{X}_{\mathrm{P}}}$ | 81.227 | **69.434** | 79.449 | 74.480 | 79.213 |

**Classifiers Trained on COCO**

| | [AUC in %] | DAVIS | ADE20k | MOSE | COCO | SA-V |
|---|---|---|---|---|---|---|
| USAM | $\Delta_{\mathcal{X}_{\mathrm{P}}}$ | 81.594 | 68.903 | 79.780 | 75.001 | 79.626 |
| USAM | $\Delta^*_{\mathcal{X}_{\mathrm{P}}}$ | 81.610 | **69.081** | **79.793** | **75.039** | **79.656** |

**Classifiers Trained on SA-V**

| | [AUC in %] | DAVIS | ADE20k | MOSE | COCO | SA-V |
|---|---|---|---|---|---|---|
| USAM | $\Delta_{\mathcal{X}_{\mathrm{P}}}$ | 81.507 | 68.769 | 79.741 | 74.817 | 79.734 |
| USAM | $\Delta^*_{\mathcal{X}_{\mathrm{P}}}$ | 81.488 | 68.832 | 79.723 | **74.824** | **79.788** |

*Table 13.* Task uncertainty quantification. The table shows the area under curve (AUC) when predicting a variable fraction of the most uncertain samples with the correct task, otherwise with the one selected by the SamScore. The most uncertain samples are determined by the respective method. Supplementary data to Table 3. Compared to Table 3, this table shows absolute AUC. To help interpreting the values, the best (*Oracle*) and worst (*Worst*) possible correction order based on the uncertainty score is added.

**Standard Measures**

| | [AUC in %] | DAVIS | ADE20k | MOSE | COCO | SA-V |
|---|---|---|---|---|---|---|
| | Oracle | 74.583 | 59.938 | 76.687 | 70.635 | 76.688 |
| | Worst | 60.567 | 51.860 | 70.400 | 61.273 | 66.643 |
| SAM | SamScore | 70.149 | 57.216 | 74.837 | 67.273 | 72.556 |
| SAM | $H_{\mathrm{Std}}$ | 70.205 | **58.039** | 75.024 | **68.253** | 73.762 |
| Bayes | $H_{\mathcal{Y}}$ | **71.063** | 57.073 | 74.633 | 67.650 | 73.724 |
| Bayes | $H_{\Theta}$ | 69.870 | 57.827 | 74.956 | 67.916 | 73.517 |
| Bayes | $H_{\mathcal{A}}$ | 66.715 | 56.125 | 74.552 | 66.661 | **74.490** |
| Bayes | $H_{\mathcal{X}_{\mathrm{P}}}$ | 69.112 | 56.870 | 74.810 | 67.678 | 73.524 |

**Classifiers Trained on DAVIS**

| | [AUC in %] | DAVIS | ADE20k | MOSE | COCO | SA-V |
|---|---|---|---|---|---|---|
| USAM | $\Delta_{\mathcal{A}}$ | 73.749 | **58.506** | 75.625 | **69.338** | 73.131 |
| USAM | $\Delta^*_{\mathcal{A}}$ | **73.785** | 58.289 | 75.307 | 69.096 | 72.860 |

**Classifiers Trained on ADE20k**

| | [AUC in %] | DAVIS | ADE20k | MOSE | COCO | SA-V |
|---|---|---|---|---|---|---|
| USAM | $\Delta_{\mathcal{A}}$ | **73.682** | **59.379** | **76.223** | **69.916** | **75.767** |
| USAM | $\Delta^*_{\mathcal{A}}$ | 73.581 | 59.323 | 75.968 | 69.712 | 75.458 |

**Classifiers Trained on COCO**

| | [AUC in %] | DAVIS | ADE20k | MOSE | COCO | SA-V |
|---|---|---|---|---|---|---|
| USAM | $\Delta_{\mathcal{A}}$ | 73.958 | **59.274** | **76.323** | 70.152 | 75.946 |
| USAM | $\Delta^*_{\mathcal{A}}$ | **74.135** | 59.268 | 76.311 | **70.155** | **75.970** |

**Classifiers Trained on SA-V**

| | [AUC in %] | DAVIS | ADE20k | MOSE | COCO | SA-V |
|---|---|---|---|---|---|---|
| USAM | $\Delta_{\mathcal{A}}$ | 73.680 | **59.165** | **76.304** | **70.009** | 76.102 |
| USAM | $\Delta^*_{\mathcal{A}}$ | **73.842** | 59.104 | 76.266 | 69.984 | **76.146** |

*Table 14.* Uncertainty quantification. The table shows the area under curve (AUC) when correcting a variable fraction of the most uncertain samples. The most uncertain samples are determined by the respective method. Oracle denotes an optimal uncertainty quantification. Supplementary material to Table 4 in the main paper. Compared to Table 4, this table shows absolute AUC. To help interpreting the values, the best (*Oracle*) and worst (*Worst*) possible correction order based on the uncertainty score is added.

**Model *Tiny***

| | $\left[\text{AUC in \%}\right]$ | DAVIS | ADE20k | MOSE | COCO | SA-V |
|---|---|---|---|---|---|---|
| SAM | Oracle | 94.621 | 88.322 | 93.696 | 91.501 | 94.473 |
| | Worst | 83.036 | 72.780 | 83.647 | 80.491 | 83.564 |
| | *SamScore* | 92.274 | 84.695 | 92.366 | 88.897 | 92.482 |
| | $H_{\text{Std}}$ | 92.353 | 83.647 | 91.614 | 88.309 | 92.355 |
| Bayes | $H_{\mathcal{Y}}$ | 87.783 | 80.166 | 89.724 | 85.727 | 89.983 |
| | $H_{\Theta}$ | 93.349 | 84.693 | 92.638 | 89.597 | **93.256** |
| | $H_{\mathcal{X}_{\text{P}}}$ | 92.773 | 85.433 | 92.218 | 88.975 | 92.740 |
| | $H_{\mathcal{A}}$ | 89.219 | 80.688 | 90.258 | 86.697 | 90.441 |
| | $\text{USAM}_{\text{T}}$ | **93.665** | **87.074** | **92.852** | **90.322** | 93.222 |

**Model *Small***

| | $\left[\text{AUC in \%}\right]$ | DAVIS | ADE20k | MOSE | COCO | SA-V |
|---|---|---|---|---|---|---|
| SAM | Oracle | 95.087 | 88.684 | 94.182 | 91.878 | 94.718 |
| | Worst | 83.816 | 73.101 | 84.491 | 81.022 | 84.064 |
| | *SamScore* | 92.935 | 84.967 | 92.900 | 89.277 | 92.885 |
| | $H_{\text{Std}}$ | 93.071 | 84.068 | 92.305 | 88.889 | 92.772 |
| Bayes | $H_{\mathcal{Y}}$ | 88.348 | 80.462 | 90.299 | 86.147 | 90.320 |
| | $H_{\Theta}$ | 93.768 | 84.949 | 93.064 | 89.915 | 93.489 |
| | $H_{\mathcal{X}_{\text{P}}}$ | 93.326 | 85.857 | 92.764 | 89.436 | 93.066 |
| | $H_{\mathcal{A}}$ | 89.808 | 81.145 | 90.803 | 87.156 | 90.761 |
| | $\text{USAM}_{\text{S}}$ | **94.143** | **87.489** | **93.431** | **90.793** | **93.608** |

**Model *Base+***

| | $\left[\text{AUC in \%}\right]$ | DAVIS | ADE20k | MOSE | COCO | SA-V |
|---|---|---|---|---|---|---|
| SAM | Oracle | 95.839 | 89.173 | 94.896 | 92.401 | 95.119 |
| | Worst | 85.224 | 73.586 | 85.881 | 81.766 | 84.941 |
| | *SamScore* | 93.502 | 85.189 | 93.786 | 89.372 | 93.247 |
| | $H_{\text{Std}}$ | 93.743 | 84.395 | 93.225 | 89.311 | 93.231 |
| Bayes | $H_{\mathcal{Y}}$ | 89.380 | 80.819 | 91.125 | 86.757 | 90.863 |
| | $H_{\Theta}$ | 94.361 | 85.225 | 93.650 | 90.313 | 93.813 |
| | $H_{\mathcal{X}_{\text{P}}}$ | 94.313 | 86.634 | 93.680 | 89.938 | 93.584 |
| | $H_{\mathcal{A}}$ | 90.407 | 81.868 | 91.736 | 87.642 | 91.282 |
| | $\text{USAM}_{\text{B}}$ | **94.994** | **87.998** | **94.200** | **91.328** | **93.981** |

**Model *Large***

| | $\left[\text{AUC in \%}\right]$ | DAVIS | ADE20k | MOSE | COCO | SA-V |
|---|---|---|---|---|---|---|
| SAM | Oracle | 96.291 | 89.361 | 95.394 | 92.735 | 95.407 |
| | Worst | 86.336 | 73.896 | 87.080 | 82.473 | 85.595 |
| | *SamScore* | 94.635 | 85.556 | 94.519 | 90.641 | 93.523 |
| | $H_{\text{Std}}$ | 94.500 | 84.757 | 93.905 | 90.095 | 93.405 |
| Bayes | $H_{\mathcal{Y}}$ | 90.198 | 81.144 | 91.784 | 87.128 | 91.219 |
| | $H_{\Theta}$ | 94.722 | 85.400 | 94.065 | 90.504 | 94.023 |
| | $H_{\mathcal{X}_{\text{P}}}$ | 95.013 | 86.854 | 94.249 | 90.593 | 93.762 |
| | $H_{\mathcal{A}}$ | 91.536 | 82.045 | 92.572 | 88.318 | 91.533 |
| | $\text{USAM}_{\text{L}}$ | **95.454** | **88.217** | **94.793** | **91.710** | **94.365** |

| | $\text{IoU}_{\text{GT}}$ | SamS | $H_{\text{Std}}$ | $H_{\mathcal{Y}}$ | $H_{\Theta}$ | $H_{\mathcal{A}}$ | $H_{\mathcal{X}_{\text{P}}}$ | $\text{USAM}_{\text{T}}$ | $\Delta_{\Theta}^{*}$ | $\Delta_{\mathcal{A}}^{*}$ | $\Delta_{\mathcal{X}_{\text{P}}}^{*}$ |
|---|---|---|---|---|---|---|---|---|---|---|---|
| $\text{IoU}_{\text{GT}}$ | 1.00 | 0.46 | -0.39 | 0.06 | -0.63 | -0.12 | -0.53 | 0.76 | -0.37 | -0.20 | -0.25 |
| SamS | 0.46 | 1.00 | -0.63 | -0.22 | -0.68 | -0.31 | -0.54 | 0.63 | -0.56 | -0.82 | -0.36 |
| $H_{\text{Std}}$ | -0.39 | -0.63 | 1.00 | 0.41 | 0.81 | 0.36 | 0.74 | -0.48 | 0.69 | 0.49 | 0.66 |
| $H_{\mathcal{Y}}$ | 0.06 | -0.22 | 0.41 | 1.00 | 0.26 | 0.42 | 0.39 | 0.11 | 0.23 | 0.33 | 0.27 |
| $H_{\Theta}$ | -0.63 | -0.68 | 0.81 | 0.26 | 1.00 | 0.33 | 0.74 | -0.66 | 0.62 | 0.45 | 0.51 |
| $H_{\mathcal{A}}$ | -0.12 | -0.31 | 0.36 | 0.42 | 0.33 | 1.00 | 0.36 | -0.15 | 0.31 | 0.17 | 0.17 |
| $H_{\mathcal{X}_{\text{P}}}$ | -0.53 | -0.54 | 0.74 | 0.39 | 0.74 | 0.36 | 1.00 | -0.51 | 0.61 | 0.39 | 0.53 |
| $\text{USAM}_{\text{T}}$ | 0.76 | 0.63 | -0.48 | 0.11 | -0.66 | -0.15 | -0.51 | 1.00 | -0.52 | -0.27 | -0.35 |
| $\Delta_{\Theta}^{*}$ | -0.37 | -0.56 | 0.69 | 0.23 | 0.62 | 0.31 | 0.61 | -0.52 | 1.00 | 0.35 | 0.72 |
| $\Delta_{\mathcal{A}}^{*}$ | -0.20 | -0.82 | 0.49 | 0.33 | 0.45 | 0.17 | 0.39 | -0.27 | 0.35 | 1.00 | 0.29 |
| $\Delta_{\mathcal{X}_{\text{P}}}^{*}$ | -0.25 | -0.36 | 0.66 | 0.27 | 0.51 | 0.17 | 0.53 | -0.35 | 0.72 | 0.29 | 1.00 |

*Table 15.* Correlation between different UQ measures on the COCO validation dataset using by the *Tiny* SAM model. $\text{IoU}_{\text{GT}}$ denotes the real intersection over union between SAMs prediction and the ground truth. Supplementary data to Table 5.

| | $\text{IoU}_{\text{GT}}$ | SamS | $H_{\text{Std}}$ | $H_{\mathcal{Y}}$ | $H_{\Theta}$ | $H_{\mathcal{A}}$ | $H_{\mathcal{X}_{\text{P}}}$ | $\text{USAM}_{\text{S}}$ | $\Delta_{\Theta}^{*}$ | $\Delta_{\mathcal{A}}^{*}$ | $\Delta_{\mathcal{X}_{\text{P}}}^{*}$ |
|---|---|---|---|---|---|---|---|---|---|---|---|
| $\text{IoU}_{\text{GT}}$ | 1.00 | 0.45 | -0.38 | 0.07 | -0.62 | -0.12 | -0.50 | 0.74 | -0.34 | -0.20 | -0.23 |
| SamS | 0.45 | 1.00 | -0.56 | -0.22 | -0.68 | -0.27 | -0.50 | 0.57 | -0.49 | -0.69 | -0.33 |
| $H_{\text{Std}}$ | -0.38 | -0.56 | 1.00 | 0.41 | 0.81 | 0.36 | 0.74 | -0.48 | 0.69 | 0.49 | 0.66 |
| $H_{\mathcal{Y}}$ | 0.07 | -0.22 | 0.41 | 1.00 | 0.26 | 0.42 | 0.39 | 0.12 | 0.23 | 0.33 | 0.27 |
| $H_{\Theta}$ | -0.62 | -0.68 | 0.81 | 0.26 | 1.00 | 0.33 | 0.74 | -0.66 | 0.62 | 0.45 | 0.51 |
| $H_{\mathcal{A}}$ | -0.12 | -0.27 | 0.36 | 0.42 | 0.33 | 1.00 | 0.36 | -0.15 | 0.31 | 0.17 | 0.17 |
| $H_{\mathcal{X}_{\text{P}}}$ | -0.50 | -0.50 | 0.74 | 0.39 | 0.74 | 0.36 | 1.00 | -0.50 | 0.61 | 0.39 | 0.53 |
| $\text{USAM}_{\text{S}}$ | 0.74 | 0.57 | -0.48 | 0.12 | -0.66 | -0.15 | -0.50 | 1.00 | -0.48 | -0.28 | -0.32 |
| $\Delta_{\Theta}^{*}$ | -0.34 | -0.49 | 0.69 | 0.23 | 0.62 | 0.31 | 0.61 | -0.48 | 1.00 | 0.35 | 0.72 |
| $\Delta_{\mathcal{A}}^{*}$ | -0.20 | -0.69 | 0.49 | 0.33 | 0.45 | 0.17 | 0.39 | -0.28 | 0.35 | 1.00 | 0.29 |
| $\Delta_{\mathcal{X}_{\text{P}}}^{*}$ | -0.23 | -0.33 | 0.66 | 0.27 | 0.51 | 0.17 | 0.53 | -0.32 | 0.72 | 0.29 | 1.00 |

*Table 16.* Correlation between different UQ measures on the COCO validation dataset using by the *Small* SAM model. $\text{IoU}_{\text{GT}}$ denotes the real intersection over union between SAMs prediction and the ground truth. Supplementary data to Table 5.

| | $\text{IoU}_{\text{GT}}$ | SamS | $H_{\text{Std}}$ | $H_{\mathcal{Y}}$ | $H_{\Theta}$ | $H_{\mathcal{A}}$ | $H_{\mathcal{X}_{\text{P}}}$ | $\text{USAM}_{\text{B+}}$ | $\Delta_{\Theta}^{*}$ | $\Delta_{\mathcal{A}}^{*}$ | $\Delta_{\mathcal{X}_{\text{P}}}^{*}$ |
|---|---|---|---|---|---|---|---|---|---|---|---|
| $\text{IoU}_{\text{GT}}$ | 1.00 | 0.35 | -0.35 | 0.07 | -0.59 | -0.11 | -0.46 | 0.72 | -0.31 | -0.20 | -0.20 |
| SamS | 0.35 | 1.00 | -0.52 | -0.29 | -0.61 | -0.27 | -0.45 | 0.42 | -0.42 | -0.66 | -0.32 |
| $H_{\text{Std}}$ | -0.35 | -0.52 | 1.00 | 0.41 | 0.81 | 0.36 | 0.74 | -0.45 | 0.69 | 0.49 | 0.66 |
| $H_{\mathcal{Y}}$ | 0.07 | -0.29 | 0.41 | 1.00 | 0.26 | 0.42 | 0.39 | 0.12 | 0.23 | 0.33 | 0.27 |
| $H_{\Theta}$ | -0.59 | -0.61 | 0.81 | 0.26 | 1.00 | 0.33 | 0.74 | -0.64 | 0.62 | 0.45 | 0.51 |
| $H_{\mathcal{A}}$ | -0.11 | -0.27 | 0.36 | 0.42 | 0.33 | 1.00 | 0.36 | -0.14 | 0.31 | 0.17 | 0.17 |
| $H_{\mathcal{X}_{\text{P}}}$ | -0.46 | -0.45 | 0.74 | 0.39 | 0.74 | 0.36 | 1.00 | -0.48 | 0.61 | 0.39 | 0.53 |
| $\text{USAM}_{\text{B+}}$ | 0.72 | 0.42 | -0.45 | 0.12 | -0.64 | -0.14 | -0.48 | 1.00 | -0.44 | -0.28 | -0.29 |
| $\Delta_{\Theta}^{*}$ | -0.31 | -0.42 | 0.69 | 0.23 | 0.62 | 0.31 | 0.61 | -0.44 | 1.00 | 0.35 | 0.72 |
| $\Delta_{\mathcal{A}}^{*}$ | -0.20 | -0.66 | 0.49 | 0.33 | 0.45 | 0.17 | 0.39 | -0.28 | 0.35 | 1.00 | 0.29 |
| $\Delta_{\mathcal{X}_{\text{P}}}^{*}$ | -0.20 | -0.32 | 0.66 | 0.27 | 0.51 | 0.17 | 0.53 | -0.29 | 0.72 | 0.29 | 1.00 |

*Table 17.* Correlation between different UQ measures on the COCO validation dataset using by the *Base+* SAM model. $\text{IoU}_{\text{GT}}$ denotes the real intersection over union between SAMs prediction and the ground truth. Supplementary data to Table 5.

