# OpenReview forum: "UncertainSAM: Fast and Efficient Uncertainty Quantification of the Segment Anything Model"
_ICML.cc/2025/Conference — ICML 2025 poster_

### Official Review · Reviewer_pXDa · 2025-03-10

**Overall Recommendation:** 3

**Summary:**

This work introduces a method for uncertainty quantification of SAM, based on Bayesian entropy formulation. A lightweight post-hoc UQ method is trained based on the formulation. Results on multiple public benchmarks demonstrate the effectiveness of the proposed method.

## update after rebuttal
I am glad to keep my original positive rating.

**Claims And Evidence:**

Claims are clear and convincing.

**Essential References Not Discussed:**

No.

**Experimental Designs Or Analyses:**

The experimental designs and analyses are reasonable to me.

**Methods And Evaluation Criteria:**

Evaluation criteria make sense.

**Other Comments Or Suggestions:**

N/A.

**Other Strengths And Weaknesses:**

Strength: The experiments on multiple public benchmarks validate the effectiveness of the proposed method comprehensively.

**Questions For Authors:**

N/A.

**Relation To Broader Scientific Literature:**

This uncertainty quantification method of SAM can benefit a wide range of computer vision tasks. Considering the wide application of SAM, this method will be influential.

**Theoretical Claims:**

Theoretical claims seem correct to me.

---

> ### Author Rebuttal · Authors · 2025-03-27
>
> We thank the reviewer for the feedback and positive assessment.
>
> To further support the reviewer’s inclination toward acceptance, we would like to highlight the linked code repository in the main paper, which demonstrates the ease of use of our proposed framework. We remain open to any additional questions, remarks, or suggestions to refine our proposed paper during the author-reviewer discussion.

---

### Official Review · Reviewer_3EoJ · 2025-03-11

**Overall Recommendation:** 3

**Summary:**

This paper introduces an interesting method to measure the uncertainty of SAM in image segmentation tasks. To achieve this, this paper proposes USAM, an efficient post-hoc method that can quantify the uncertainty of SAM and help users determine whether the model results are reliable, which tunes a lightweight MLP-based estimator. Specifically, the authors design a formula based on Bayesian entropy that can simultaneously consider three levels of uncertainty: model uncertainty (for example, the model is too small, resulting in inaccurate predictions), prompt uncertainty like ambiguous user prompts, and task uncertainty like unclear user prompts. The paper demonstrates the effectiveness of USAM on several datasets, showing its ability to identify sources of uncertainty and improve model predictions. Overall, this paper is well-organized and theoretically grounded.

**Claims And Evidence:**

The main claims about uncertainty quantification in SAM are supported by clear mathematical formulation, method, proofs, and experiments. The minor concern is the claim that "the ambiguous nature of the class-agnostic foundation model SAM challenges current uncertainty quantification approaches." is not well justified. As SAM can generate ambiguous predictions (Top-K), it inherently has uncertainty awareness, making the uncertainty analysis easier. Why do the authors claim that this will challenge UQ?

**Essential References Not Discussed:**

The proposed method explores the uncertainty quantification of the SAM model. However, this is not a new direction, and there are many related works. Even though the proposed method is comprehensive, technically sound, and theoretically correct, it is suggested to make necessary discussion with related UQ methods using SAM [1-2]

[1] Flaws can be Applause: Unleashing Potential of Segmenting Ambiguous Objects in SAM (NeurIPS 24)
[2] Uncertainty-aware Fine-tuning of Segmentation Foundation Models (NeurIPS 24)

**Experimental Designs Or Analyses:**

Most of the experiments were well conducted, especially for the detailed justification of claimed uncertainty terms from the model, prompt, and task aspects. The experimental analysis is also comprehensive and convincing. However, there are two major concerns. The first concern is insufficiently comparing existing UQ methods in SAM model, such as [1-3]. The second concern is the lack of comparison of specific settings focusing on uncertain predictions, like ambiguous segmentation, which is critical to highlight the strength of the proposed UQ methods [1,2].

[1] Flaws can be Applause: Unleashing Potential of Segmenting Ambiguous Objects in SAM (NeurIPS 24)
[2] Uncertainty-aware Fine-tuning of Segmentation Foundation Models (NeurIPS 24)

**Methods And Evaluation Criteria:**

The proposed method introduces Bayesian Entropy Approximation in SAM, which makes sense and is technically sound. It considers uncertainty from different levels (Eq.2), which is well justified by both theoretical proofs (Sec. 3) and experiments (Sec.4.2 to Sec. 4.4).

**Other Comments Or Suggestions:**

N/

**Other Strengths And Weaknesses:**

Strength:
- The paper is well organized and gives a comprehensive and theoretically sound analysis of the uncertainty in SAM.
- The proposed Bayesian approximation is reasonable.
- The perception of the model, prompt, and task uncertainty is interesting and comprehensive.
- The experimental justification is solid.

Weakness:
- Lack some comparison with related works about modelling uncertainty in SAM
- Lack some clarification (see Questions For Authors)

**Questions For Authors:**

- The authors claim that "For several tasks, quantifying the uncertainty of SAM is of particular interest.". It is suggested to clarify what tasks require measuring uncertainty in SAM.
- The authors are suggested to give a justification for the efficiency of the proposed method.
- In Sec. 4, the authors implement the method in SAM2, but Fig.1 illustrates SAM, which is inconsistent.
- Considering that the authors use SAM 2 as the base model, the theoretical analysis lacks a formulation for time-series signals. Will it introduce extra uncertainty when considering the object motions in videos?
- Lack of a comparison on the ambiguous segmentation benchmark.

**Relation To Broader Scientific Literature:**

The proposed method has a potentially broader impact on understanding the existing SAM model from the uncertainty perspective, thereby inspiring the algorithmic designs of the next-generation SAM.

**Theoretical Claims:**

I have checked the theoretical claims about the uncertainty quantification formulation (Sec. 2.2), the grounding in SAM (Sec.2.3), and Bayesian Entropy Approximation (Sec.3 and Appendix A). I did not find major theoretical errors and think that this paper is well organized.

---

> ### Author Rebuttal · Authors · 2025-03-27
>
> We acknowledge the in-depth review that recognizes our strengths and provides valuable directions for addressing flaws, particularly regarding related methods. Below, we address questions and concerns. Issues already covered in other responses are only referenced.
>
> ---
>
> ## Claims
>
> Thank you for pointing out that this claim in the abstract is not sufficiently elaborated in the introduction. A more detailed discussion of SAM and its implications for UQ is provided in Section 2.3, "UQ in SAM".
> Most UQ methods are based on MC strategies and motivated by Bayesian principles (see Equation 1 ff.). While these methods work well in practice, their theoretical justifications rely on assumptions that are often unrealistic and cannot be guaranteed. For example, test-time augmentation (Wang et al., 2019), used to estimate aleatoric uncertainty, assumes that the sample is in-distribution. Similarly, epistemic UQ estimators, such as model ensembles, assume that uncertain samples are out-of-distribution. These issues, which already exist in well-established UQ approaches, are further exacerbated by the task-agnostic nature of SAM.
> To the best of our knowledge, there is no general discussion in the literature on SAM's implications on UQ. However, we acknowledge that our formulation of this claim is unclear. To address this, we suggest adding the following to line 126 (right column):
>
> "While the presented methods work well in practice, the theoretical foundations for separating aleatoric and epistemic uncertainty remain debatable and under active discussion (Gruber et al., 2023). The task-agnostic nature of SAM further amplifies this challenge, as discussed next. More..."
>
> ## Weakness 1
>
> As the reviewer mentioned, several papers propose UQ methods, such as [1] and [2]. The first encodes variance into the latent to sample ambiguous segmentations, while [2] estimates pixel-wise epistemic uncertainty to improve finetuning. They are designed for specific tasks and not directly comparable to us. USAM neither samples mask nor estimates pixel-wise uncertainty, making it unsuitable for both.
>
> However, our UQ comparison already aligns with the comparisons performed in [1] and [2].
> In [1], UQ performance is evaluated by applying prompt perturbations (similar to our H_XP)  to SegGPT and image augmentations (similar to our H_Y) to SEEM. Note that by changing the frameworks to SegGPT and SEEM, the ambiguous segmentation task is evaluated but UQ is not evaluated in isolation. In [2], a pixel-wise equivalent to our "Entropy" is used, termed SUM Confidence-Uncer, alongside a pixel-wise equivalent to our H_A, termed SUM Discrepancy-Uncer. These study designs underline that standard UQ methods are suitable for evaluating new UQ approaches.
> Nevertheless, we acknowledge that our paper does not sufficiently address the variety of UQ approaches used in conjunction with SAM. We propose to expand Section 2.3 ("UQ in SAM") and include a discussion of more related works that propose or employ UQ with SAM, such as [1] and [2].
>
> ---
>
> ## Q1
>  "For several tasks, quantifying the uncertainty of SAM is of particular interest". We mention some tasks in the proposed paper: For example, Vujasinovic et al. (2024) use UQ to estimate if supervision is required to correct tracking errors, or in the healthcare domain to gain trust and improve prediction (Deng et al., 2023; Zhang et al., 2023). The papers [1] and [2] mentioned by the reviewer employ UQ in SAM to sample prediction or to improve finetuning. We will add these additional references to the camera ready.
>
> ---
>
> ## Q2
>
> A similar concern was raised by reviewer **nnaG**. We agree that the simplicity of our MLP does not sufficiently support claims of superior efficiency. To address this, we conducted runtime experiments demonstrating that USAM incurs minimal computational overhead and is significantly faster than MC-based methods.
> For details, see our response to reviewer **nnaG**, Weakness 3, or run the experiment in our anonymous repository https://anonymous.4open.science/r/UncertainSAM-C5BF/scripts/speed_test.ipynb. We propose adding these findings to the camera-ready version.
>
> ---
>
> ## Q3 and Q4
>
> Our submitted paper contains ambiguities regarding our use of SAM vs. SAM2. For a detailed clarification, please see our introduction of the response to reviewer **6x8E**.
>
> ---
>
> ## Q5
>
> We are uncertain how our method could be applied to the ambiguous segmentation benchmark. As stated by [X], the goal of this benchmark is to achieve "better agreement between prediction and the ground truth".
> In contrast, our method estimates the variance of the distribution (i.e., uncertainty). Since we do not modify SAM’s mask prediction, USAM’s segmentations would be identical to vanilla SAM’s.
>
> [X] Rahman, Aimon, et al. "Ambiguous medical image segmentation using diffusion models." CVPR. 2023.

---

> > ### Comment · Reviewer_3EoJ · 2025-04-05
> >
> > Thanks for the rebuttal, which addresses my concerns. Hence, I am happy to keep my score.

---

### Official Review · Reviewer_6x8E · 2025-03-14

**Overall Recommendation:** 3

**Summary:**

This paper makes a series of efforts to enable uncertainty quantification for SAM. To achieve this, the authors first adopt Monte Carlo sampling to estimate the predictive, epistemic model, aleatoric prompt, and aleatoric task uncertainty. Then, to release the computation burden of the sampling process during testing, the authors concatenate SAMs 256-dimensional mask and IoU tokens and feed them into MLPs that are trained to predict uncertainty.

**Claims And Evidence:**

The claims are well-supported by the experiments.

**Essential References Not Discussed:**

No

**Experimental Designs Or Analyses:**

The experimental designs are reasonable with no major issues. One minor issue is the lack of details on how to
combine SAM and SAM2 for the evaluation.

**Methods And Evaluation Criteria:**

The proposed method is reasonable.

**Other Comments Or Suggestions:**

No other comments.

**Other Strengths And Weaknesses:**

Strengths
1. Uncertainty quantification is important for segmentation tasks.
2. The decomposition of aleatoric uncertainty into prompt and task uncertainty is reasonable.
3. The proposed method is easy to follow.

Weaknesses
1. Lacks the analysis of why the mask and IoU tokens can reflect the uncertainty. Since the uncertainty depends on the ground truth mask, the mask and IoU tokens can not reflect uncertainty entirely.

2. Why choose MLPs and how to confirm their specific parameters e.g., 3 Layers with 512 hidden states and a Sigmoid activation.

**Questions For Authors:**

Since the uncertainty depends on the ground truth mask, the mask and IoU tokens can not reflect uncertainty entirely. Thus, it remains questionable that the usage of only the mask and IoU tokens can reflect the uncertainty.

**Relation To Broader Scientific Literature:**

This work extends the existing Uncertainty Quantification into SAMs.

**Theoretical Claims:**

There are no new theoretical proofs.

---

> ### Author Rebuttal · Authors · 2025-03-27
>
> We want to thank the reviewer for highlighting our strenghts and asking questions that are valuable for eliminating weaknesses and enhancing our paper in terms of readability and evaluation.
>
> ---
>
> Before addressing the questions, we need to clarify an issue that affects this review as well as reviewer **3EoJ**.
> After carefully re-reading the paper with temporal distance, we acknowledge that our description of the usage of SAM2 is misleading and needs to be improved:
>
> - First, line 254 (right column), "We combine the evaluation of SAM and SAM2," is misleading because we do not combine the models during evaluation. Rather, we combine the datasets evaluated in the corresponding papers. We do this to provide a broader overview that also includes the latest data publication from SAM2.
> - Second, line 242 (right column), "We use the pretrained SAM2," suggests that we use the video segmenation mechanism of SAM2, which is designed for segmenting temporally consistent masks in videos. In reality, while we use the pretrained SAM2 model, we only use its single-image implementation, which is equivalent to SAM and does not include the video extension. We do this because the authors of SAM2 provide more stable code, better interfaces, and slightly better accuracy.
>
> Potentially, our MLPs could be used ad hoc to estimate uncertainty from the refined tokens of SAM2 applied to videos, but that is not evaluated and is beyond the scope of this paper. To prevent misunderstandings for future readers, we will avoid the term SAM2 and suggest the following modifications in the camera-ready version:
>
>
> **Replace:** "In our experiments, we use the pretrained SAM2 (Ravi et al., 2024)" **with** "In our experiments, we use SAM models pretrained by Ravi et al. (2024)."
>
> **Replace:** "We combine the evaluation of SAM and SAM2 ..." **with** "Combining the dataset selection of Kirillov et al. and Ravi et al., we ..."
>
> ---
>
> ## Weakness and Question 1: Why the mask and IoU tokens can reflect uncertainty
>
> (Un)certainty refers to the likelihood that the prediction is correct. In our case, correctness is measured by the IoU metric, making the expected IoU an ideal proxy for uncertainty. In general, estimating uncertainty does not require ground truth. For example, predictive uncertainty is often described as the variance of the predictive distribution. As seen in Equations 1 and 2, this distribution does not depend on the ground truth. Therefore, the absence of ground truth during inference does not invalidate the expressiveness of our method that uses tokens for uncertainty estimation.
>
> Additionally, the SamScore in SAM (Kirillov et al.) is based on the same principle and also does not have access to ground truth during inference. However, the question of whether and how the mask and IoU tokens reflect uncertainty is an interesting one:
>
> Essentially, the MLPs attempt to map patterns from the object’s latent space (i.e., the tokens) to the expected IoU gap (i.e., uncertainty) based on statistical evidence. The already given effectiveness of the SamScore suggests that such patterns exist in general.
>
> To further investigate and provide more details for future readers, we conducted an additional ablation study. In this ablation, we set either the IoU token or the mask token to 0 to remove potentially informative patterns.
> We evaluated using the tiny model across the three tasks described in the submitted paper on the COCO dataset. The results indicate that both tokens contribute to the predictive capability, yet they remain accurate as standalone features. Both in combination lead to the best results. The results are:
>
>
> **Token to Uncertainty Ablation** **in relative AUC in %**
>
> | **Mask Token** | **IoU Token** | **Model UQ** | **Prompt UQ** | **Task UQ** |
> |:---:|:---:|:---:|:---:|:---:|
> | No | Yes | 61.19 | 72.08 | 86.89 |
> | Yes | No | 62.63 | 76.42 | 91.96 |
> | Yes | Yes | **63.66** | **78.30** | **94.82** |
>
> We will add this interesting findings to the camera ready version.
>
> ## Weakness 2: The MLPs' design choice
>
> The reviewer’s remark about our MLP design choice is similar to the comment from reviewer **nnaG**, indicating that our current explanation lacks clarity. There was no dedicated architecture design process for the UQ estimator. Instead, the architecture is motivated by SAM’s original MLP design for predicting the SamScore. We differ only in the input dimension and hidden states (increasing from 256 to 512) because we use and concatenate both tokens instead of using just one.
>
> The sigmoid activation function is a reasonable choice since the expected IoU gap ranges between 0 and 1. To improve clarity, we suggest modifying the method description and adding a figure to illustrate the MLP architecture. Please see our response to reviewer **nnaG** for details on the suggested changes.

---

### Official Review · Reviewer_nnaG · 2025-03-18

**Overall Recommendation:** 3

**Summary:**

The paper introduces UncertainSAM (USAM), a method for uncertainty quantification (UQ) in the Segment Anything Model (SAM). By decomposing uncertainty into epistemic (model), aleatoric (prompt/task), and task ambiguity components, USAM employs a Bayesian entropy framework and lightweight MLPs to estimate uncertainty efficiently. Experimental results across multiple datasets demonstrate USAM's superiority over SAM's inherent confidence score and standard UQ baselines, offering a practical solution for applications requiring reliability and efficiency.

**Claims And Evidence:**

good

**Essential References Not Discussed:**

None

**Experimental Designs Or Analyses:**

good

**Methods And Evaluation Criteria:**

good

**Other Comments Or Suggestions:**

none

**Other Strengths And Weaknesses:**

Strengths:
1. It introduces a novel decomposition of uncertainty into epistemic, prompt, and task sources, addressing SAM's unique class-agnostic nature. This provides a nuanced framework for understanding SAM's limitations. WDSI effectively addresses the challenge of progressive occlusion, where parts of objects become invisible due to overlapping or changes in camera viewpoint. By leveraging hypergraph representations and the WSW metric, the framework ensures reliable tracking and segmentation even in occlusion-heavy scenes.

2. USAM’s lightweight MLPs achieve state-of-the-art performance while avoiding the computational overhead of Bayesian methods. Extensive experiments on five datasets (DAVIS, ADE20k, MOSE, COCO, SA-V) validate USAM’s effectiveness in tasks like model selection, prompt refinement, and task supervision.


Weaknesses
1. While USAM outperforms SAM’s confidence score and entropy-based methods, it does not compare against recent SAM-specific UQ approaches (e.g., prompt-augmented methods), leaving its relative novelty unclear. And, though USAM is claimed to be efficient, no runtime or memory benchmarks are provided to quantify gains over Bayesian sampling or other UQ methods. This information would help researchers understand the practical feasibility of implementing USAM.

2. The MLP architecture and SMAC3 optimization setup are underdescribed. The author should provide more structural diagrams to clarify.

3. As a lightweight method, the authors should add parameter and speed comparison.

**Questions For Authors:**

please see weakness

**Relation To Broader Scientific Literature:**

Please see weakness

**Theoretical Claims:**

good

---

> ### Author Rebuttal · Authors · 2025-03-27
>
> Thanks to the reviewer for the detailed feedback and comprehensible remarks that will help enhance the paper. Our responses to the proposed improvements are as follows:
>
> ---
>
> ## Weakness 1
>
> We only partially agree with the statement that we do not compare to SAM-specific UQ. For example, our Prompt UQ is closely related to Deng et al. (2023, see main paper for reference). Furthermore, SAM-specific UQ is usually application-specific and not readily translatable to our holistic view of UQ. However, this weakness will be addressed with a modification of the paper, which is further elaborated in our response to reviewer 3EoJ, who identified a similar concern. Please see response to reviewer 3EoJ for the details.
>
> ---
>
> ## Weakness 2
>
> We agree that the **architecture** is underdescribed. It is motivated by SAM's original MLP design for predicting the SamScore. We differ only in the input dimension and hidden state (from 256 to 512) because we employ and concatenate both tokens. We suggest the following modifications to clarify this in the paper:
>
> - Line 237: "The simple design of the MLPs aligns with the architecture of SAM’s inherent MLP for predicting the SamScore, as described in the Appendix of Kirillov et al. (2023). They have three layers and a sigmoid activation, but we use 512 input dimensions and hidden states due to the concatenated input tokens. The architecture is visualized in Figure XX."
> - In addition, we will include a figure presenting an MLP with the input tokens.
>
> Regarding **SMAC3:** Due to space limitations, we presented the upper and lower hyperparameter bounds optimized using SMAC3 in ll. 247–252 (right column). We suggest adding a more detailed description and presenting the per-iteration results in tabular form. Since the optimization is not the primary focus of the paper, we will include this in the Appendix and refer to it in line 252. The SMAC3 results will provide greater transparency regarding the robustness of the proposed method.
>
> ---
>
> ## Weakness 3
>
> Our proposed MLPs are very small compared to the entire SAM model and require only a single forward pass, unlike MC-based methods. However, as mentioned by the reviewer, an explicit performance comparison should be included in the final version of the paper to demonstrate the impact of our method on efficiency. To address this, we designed hardware-dependent runtime experiments and added them to our anonymous code repository at https://anonymous.4open.science/r/UncertainSAM-C5BF/scripts/speed_test.ipynb for reproducibility.
>
> To illustrate the practical efficiency impact, we conducted tests on a small consumer GPU (GeForce RTX 3050) using setups described in the paper. We compare us to vanilla SAM, entropy, and MC sampling (image and prompt) as employed in our paper. Notably, we performed the test with all nine (!) MLPs presented in our paper. Thus, the efficiency of our method could be increased by up to a factor of nine if only specific MLPs are necessary for a given application. It shows that our method is superior to all other extended UQ methods. We thank the reviewer for this suggestion and propose adding the following results to the camera ready:
>
>
> **Large Model**
> | Config                   | Time (s/iter) | Params          |
> |--------------------------|--------------|----------------|
> | SAM2                     | 0.43796      | 224.4M         |
> | SAM2 + Entropy           | 0.45278      | 224.4M         |
> | SAM2 + 5 MC img aug      | 2.18681      | 224.4M         |
> | SAM2 + 8 MC prompt aug   | 0.50030      | 224.4M         |
> | SAM2 + 9 MLPs **(USAM)**     | **0.44144**      | 224.4M + 9×526K |
>
> **Base+ Model**
> | Config                   | Time (s/iter) | Params          |
> |--------------------------|--------------|----------------|
> | SAM2                     | 0.20538      | 80.8M          |
> | SAM2 + Entropy           | 0.23294      | 80.8M          |
> | SAM2 + 5 MC img aug      | 1.02753      | 80.8M          |
> | SAM2 + 8 MC prompt aug   | 0.28902      | 80.8M          |
> | SAM2 + 9 MLPs **(USAM)**     | **0.20966**      | 80.8M + 9×526K |
>
> **Small Model**
> | Config                   | Time (s/iter) | Params          |
> |--------------------------|--------------|----------------|
> | SAM2                     | 0.13444      | 46.0M          |
> | SAM2 + Entropy           | 0.15716      | 46.0M          |
> | SAM2 + 5 MC img aug      | 0.68734      | 46.0M          |
> | SAM2 + 8 MC prompt aug   | 0.23210      | 46.0M          |
> | SAM2 + 9 MLPs **(USAM)**     | **0.14160**      | 46.0M + 9×526K |
>
> **Tiny Model**
> | Config                   | Time (s/iter) | Params          |
> |--------------------------|--------------|----------------|
> | SAM2                     | 0.12166      | 38.9M          |
> | SAM2 + Entropy           | 0.14876      | 38.9M          |
> | SAM2 + 5 MC img aug      | 0.58362      | 38.9M          |
> | SAM2 + 8 MC prompt aug   | 0.19736      | 38.9M          |
> | SAM2 + 9 MLPs **(USAM)**     | **0.13899**      | 38.9M + 9×526K |

---

### Decision · Program_Chairs · 2025-05-01

**Decision:**

Accept (poster)

**Comment:**

The paper received four weak accept ratings. Initially, the reviewers had a few concerns, including some technical clarifications, runtime and parameter analysis, more experiments and analysis of uncertainty. After the rebuttal, the reviewers are satisfied with the answers and thus all support the paper's acceptance. The AC took a close look at the paper/rebuttal/reviews, and agrees with the reviewers' assessment. Thus, the AC recommends the acceptance decision and encourages the authors to revise the paper following the reviewers' suggestions, as well as releasing the code/models for reproducibility.